# The Multimodal Universe:
# Enabling Large-Scale Machine Learning with 100 TB of Astronomical Scientific Data

**The Multimodal Universe Collaboration**

Eirini Angeloudi[1,2], Jeroen Audenaert[3], Micah Bowles[4,5],
Benjamin M. Boyd[6], David Chemaly[6], Brian Cherinka[7], Ioana Ciucă[8,9,10],
Miles Cranmer[6,5] Aaron Do[6], Matthew Grayling[6], Erin E. Hayes[6],
Tom Hehir[6,5] Shirley Ho[11,12,13,5], Marc Huertas-Company[1,2,9],
Kartheik G. Iyer[14,11,9], Maja Jablonska[10,9] Francois Lanusse[11,5,15],
Henry W. Leung[16], Kaisey Mandel[6], Juan Rafael Martínez-Galarza[17,18]
Peter Melchior[13], Lucas Meyer[11,5], Liam H. Parker[11,5,19], Helen Qu[20]
Jeff Shen[13], Michael J. Smith[21,9], Connor Stone[22,23,24], Mike Walmsley[16],
John F. Wu[7,25]

[1]Instituto de Astrofisica de Canarias [2]Universidad de La Laguna
[3]Massachusetts Institute of Technology [4]University of Oxford [5]Polymathic AI
[6]University of Cambridge [7]Space Telescope Science Institute
[8]Stanford University [9]UniverseTBD [10]Australian National University
[11]Flatiron Institute [12]New York University [13]Princeton University
[14]Columbia University [15]Université Paris-Saclay, Université Paris Cité, CEA,
CNRS, AIM [16]University of Toronto
[17]Center for Astrophysics, Harvard & Smithsonian [18]AstroAI
[19]University of California, Berkeley [20]University of Pennsylvania [21]Aspia Space
[22]Université de Montréal [23]Ciela Institute [24]Mila [25]Johns Hopkins University

## Abstract

We present the MULTIMODAL UNIVERSE, a large-scale multimodal dataset of scientific astronomical data, compiled specifically to facilitate machine learning research. Overall, the MULTIMODAL UNIVERSE contains hundreds of millions of astronomical observations, constituting 100 TB of multi-channel and hyper-spectral images, spectra, multivariate time series, as well as a wide variety of associated scientific measurements and "metadata". In addition, we include a range of benchmark tasks representative of standard practices for machine learning methods in astrophysics. This massive dataset will enable the development of large multi-modal models specifically targeted towards scientific applications. All codes used to compile the MULTIMODAL UNIVERSE and a description of how to access the data is available at https://github.com/MultimodalUniverse/MultimodalUniverse

## 1 Introduction

Web-scale datasets containing billions of text and image examples have been instrumental in the development and scaling of large foundation models for language and vision [125, 60]. Surprisingly, unified datasets of comparable scale have not yet been developed for many scientific domains, hindering the progress of large, advanced machine learning (ML) models for

38th Conference on Neural Information Processing Systems (NeurIPS 2024) Track on Datasets and Benchmarks.

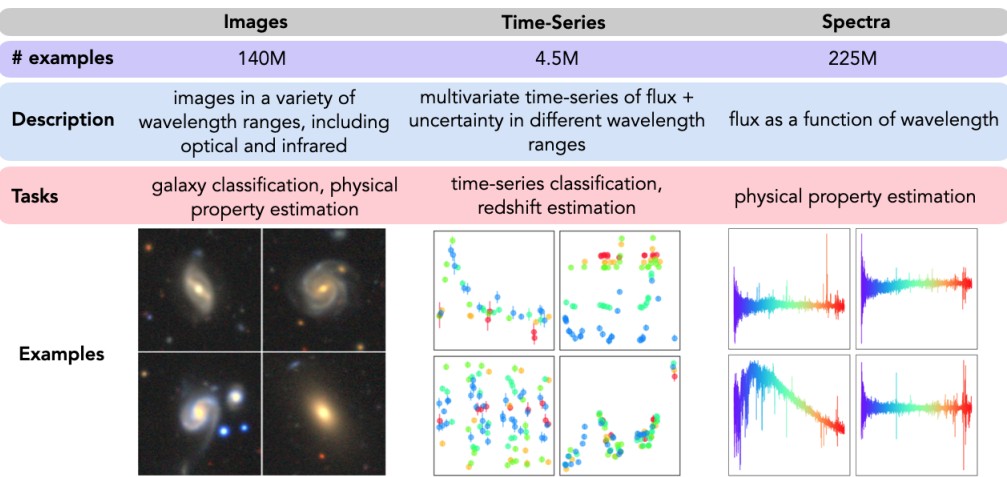

Figure 1: Illustration of the main modalities included in the MULTIMODAL UNIVERSE, along with typical associated machine learning tasks. In addition, the MULTIMODAL UNIVERSE also includes a small amount of hyperspectral images and tabular data, not shown here.

science [130]. Yet, the integration of large-scale, standardized, and comprehensive scientific datasets spanning multiple data modalities is essential for the development of ML models capable of fully leveraging the diverse spectrum of scientific data beyond traditional text and images.

To that end, we introduce the MULTIMODAL UNIVERSE (Figure 1), a large-scale, multimodal dataset of scientific data designed not only for ML research in astronomy but also to enable the broader development of scientific foundation models. The MULTIMODAL UNIVERSE capitalizes on the wide variety of astronomical observations available from a diversity of ground- and space-based telescopes to aggregate a dataset of hundreds of millions of astrophysical objects and phenomena. Altogether, this constitutes over 100 TB of open-access and copyright-free data spanning multiple observational modalities, including multi-channel and hyperspectral images, optical spectra, multivariate time-series, and an extensive array of associated scientific measurements. This diverse dataset provides an unprecedented resource for the development of sophisticated ML models for astrophysics and scientific data in general.

In keeping with its scientific goals, the MULTIMODAL UNIVERSE also emphasizes the importance of relevant scientific "metadata". This includes relevant contextual information for each observation (e.g. observational noise, pixel scale, instrumental response, etc.) that not only helps preserve the dataset's utility for scientific ML but also enables the development of models that integrate information beyond the raw data level. In addition to the dataset, we also describe a range of benchmark tasks and baseline deep learning models that reflect current best practices in astrophysics. These benchmarks provide a foundation for evaluating new models, enabling researchers to compare their approaches against established standards and push the boundaries of what is possible with ML in astronomy.

Finally, extensibility and accessibility are core to the MULTIMODAL UNIVERSE project. All data subsets are accessible as standardized Hugging Face datasets. They can be downloaded locally in full or accessed as streaming datasets directly from the Hugging Face Hub, see section 6. The code used to compile each component of the MULTIMODAL UNIVERSE from its respective official data source is hosted on the project's GitHub repository for transparency and reproducibility, and utilities to generate cross-matched (and multi-modal) versions of the dataset are also included. The architecture of the MULTIMODAL UNIVERSE source code is consistent, maintainable, and extensible. This is in contrast to the common practice in the field, which is for independent researchers, groups, or collaborations to decide their data formats in isolation of one another, creating a significant barrier to entry for multimodal research in astronomy. By making MULTIMODAL UNIVERSE publicly available, we aim to

catalyze innovation and collaboration among the astrophysics and ML communities. We believe that this dataset will not only advance our understanding of the universe but also contribute to the broader development of multimodal and metadata-aware ML methodologies.

## 2 Related Work

**Scientific Machine Learning Datasets in Other Fields** Large curated and open datasets are gaining traction outside of the textual domain. In remote sensing, Major-TOM [54] is a collated set of ESA Copernicus mission data that aims to standardise earth observation imagery into a common, ML-friendly format. Currently, Major-TOM comprises a combined 62 TB of imagery. Like the MULTIMODAL UNIVERSE, Major-TOM is run as an open source project that the community can collectively contribute to. In a similar vein, the MOMENT project [62] introduces a large time-series dataset, as a publicly-available collection of 1.2B timestamps taken from 13 domains. These domains include—but are not limited to—medical information, economic indicators, power consumption, IoT weather data, speech, and general instrumental monitoring.

**Large-Scale Machine Learning Datasets in Astrophysics** While astrophysics is inherently data-rich, most datasets have been compiled with traditional analyses in mind. This has resulted in data archives that are experiment-specific, non-uniformly stored, and not optimized for ML applications. Indeed, only a few exceptions exist where large datasets have been compiled for ML. For instance, [134, 132] introduced a substantial dataset of 76M images, enabling one of the first instances of self-supervised learning in astrophysics. Additionally, the PLAsTiCC [138] light curve classification challenge is the largest collection of astronomical time-series data for ML research, with 3.5 million simulated light curves. Finally, the Galaxy Zoo project has provided large galaxy morphology classification datasets for ML [10, 145] that have been instrumental in various astronomy specific ML studies. This includes early astronomy applications of CNNs [45], Bayesian deep learning [148], and modern neural scaling law analyses [144, 131]. However, none of these examples provide multimodal data, nor reach the scale of the data collected for the MULTIMODAL UNIVERSE.

## 3 Creating a Large-Scale Dataset of Diverse Astronomical Data

Modern astronomy has entered an era of large-scale, systematic surveys, supported by a range of space-based and ground-based instruments. These surveys generate unprecedented volumes of science-ready data products that are publicly available through open-access databases. However, these products are typically designed for conventional scientific workflows and not well-suited for ML applications; for example, astronomical imaging surveys typically provide full-frame mosaics from which individual uniform galaxy images must be cut out. As such, using these datasets effectively requires a high level of expertise and can be time-consuming. In addition, data structures and data access vary significantly among surveys, making it difficult to navigate diverse systems and interfaces to gather the necessary data. Therefore, to make use of astrophysical data, researchers must understand the intricacies of each separate survey and corresponding archive. This steep learning curve is a significant barrier to entry for ML applications in astrophysics, highlighting the need for a more streamlined and standardized approach to data access and preparation.

### 3.1 Methodology

**Data Curation Strategy** To overcome the barriers between different subfields and datasets imposed by the significant domain-specific scientific expertise required, we build the MULTIMODAL UNIVERSE by drawing on a community of diverse domain scientists. To facilitate their engagement, we have structured the data curation process to have a relatively low barrier to entry. Specifically, we enable domain scientists to contribute by simply adding scripts that download and process data from the official archives they know how to access into a standardized MULTIMODAL UNIVERSE format. While we do not impose stringent code quality requirements at this stage, it is essential that these scripts are reproducible and clearly document any selection or cuts applied to the original data. This approach

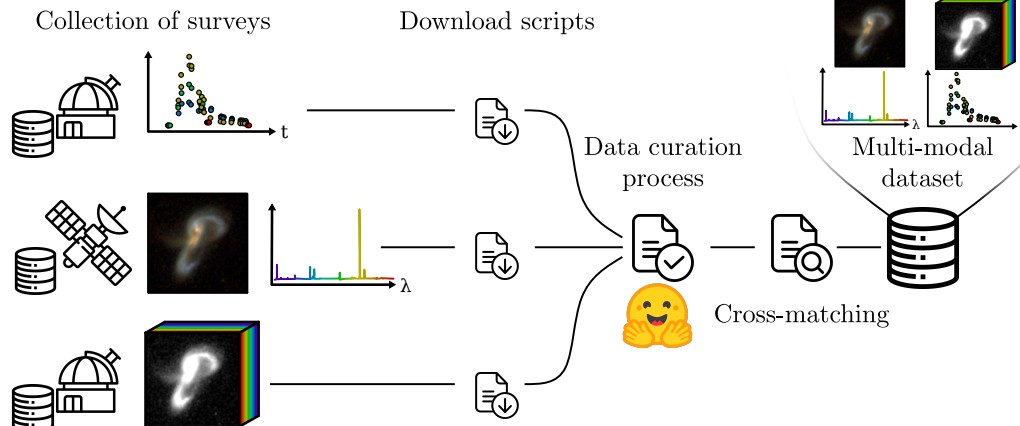

Figure 2: Illustration of the methodology behind the MULTIMODAL UNIVERSE. Domain scientists with expertise in a given astronomical survey provide data download and formatting scripts through Pull Requests on the project repository https://github.com/MultimodalUniverse/MultimodalUniverse. All datasets are then downloaded from their original source and made available as Hugging Face datasets sharing a common data schema for each modality and associated metadata. End-users can then generate any combination of MULTIMODAL UNIVERSE subsets using provided cross-matching utilities to generate multimodal datasets.

allows researchers to contribute without extensive software engineering skills, as long as data collection is transparent and well-documented. All data collection scripts are publicly available to ensure the data curation process is reproducible. Once data is downloaded from its official repository, our data curation strategy involves adopting the HuggingFace Datasets framework and imposing a standardized format for each modality, ensuring compatibility across different surveys and ease of use for ML applications.

**Multimodal Cross-Matching**  Multimodality is a crucial aspect of the MULTIMODAL UNIVERSE, but it presents several challenges. Specifically, each astronomical survey provides data in a single modality; therefore, to build multimodal examples, it is necessary to cross-match observations from several sources. Fortunately, a fundamental property of every survey is the provision of sky coordinates for each observation, enabling cross-survey matching via coordinate matching. However, the intersection between two or more surveys is often relatively small. This limitation inherently prevents the collection of dataset pairs on the scale of projects like Laion-5B [125], primarily because some modalities are much more expensive to acquire than others. Consequently, it is essential to provide access to large-scale unpaired datasets, along with the tools to easily generate various multimodal dataset pairings. The MULTIMODAL UNIVERSE addresses this need by offering a set of common cross-matched datasets and utilities that enable end-users to generate custom cross-matches tailored to their specific scientific applications. This approach ensures flexibility and maximizes the dataset's utility across different research needs.

Figure 2 summarizes the steps of our data curation and data access methodology.

## 3.2  Overview of the Dataset

Table 1 provides a breakdown of the different scientific modalities included in the MULTIMODAL UNIVERSE, along with a description of their origin and their scientific relevance. Overall, the MULTIMODAL UNIVERSE represents 100 TB of data, largely dominated by 88TB of multi-band imaging.

We provide below further description of these modalities and summarize the standardized fields and metadata defined for the main modalities of the MULTIMODAL UNIVERSE in Table 2. A complete description of the schema for each modality in the dataset is further provided in Appendix A.

Table 1: Summary of samples included in the Multimodal Universe, details of all samples are provided in [Appendix A](). $N_c$ indicates the number of channels of a given observation. [1] These are represented as 110 basis coefficients that can be resampled to an arbitrary wavelength grid. [2] Simulated dataset.

| Modality | Source Survey | $N_c$ | Shape | Number of samples | Main science |
|---|---|---|---|---|---|
| Images | Legacy Surveys DR10 [43] | 4 | 160×160 | 124M | Galaxies |
| | Legacy Surveys North [43, 134] | 3 | 152×152 | 15M | Galaxies |
| | HSC [5, 3] | 5 | 160×160 | 477K | Galaxies |
| | BTS [56, 114, 120] | 3 | 63×63 | 400K | Supernovae |
| | JWST[13, 14, 50] | 6-7 | 96×96 | 300K | Galaxies |
| Spectra | Gaia BP/RP [59] | - | 110[1] | 220M | Stars |
| | SDSS-II [1] | - | Variable | 4M | Galaxies, Stars |
| | DESI [41] | - | 7081 | 1M | Galaxies |
| | APOGEE SDSS-III [6] | - | 7514 | 716k | Stars |
| | GALAH [28] | - | Variable | 325k | Stars |
| | Chandra [51] | - | Variable | 129K | Galaxies, Stars |
| | VIPERS [126] | - | 557 | 91K | Galaxies |
| Hyperspectral Image | MaNGA SDSS-IV [2] | 4563 | 96×96 | 12k | Galaxies |
| Time Series | PLAsTiCC[2][138] | 6 | Variable | 3.5M | Time-varying objects |
| | TESS [121, 33] | 1 | Variable | 1M | Exoplanets, Stars |
| | CfA Sample [68, 69, 18, 70] | 5-11 | Variable | 1K | Supernovae |
| | YSE [7] | 6 | Variable | 2K | Supernovae |
| | PS1 SNe Ia [127] | 4 | Variable | 369 | Supernovae |
| | DES Y3 SNe Ia [24] | 4 | Variable | 248 | Supernovae |
| | SNLS [63] | 4 | Variable | 239 | Supernovae |
| | Foundation [53, 81] | 4 | Variable | 180 | Supernovae |
| | CSP SNe Ia [36, 135, 86] | 9 | Variable | 134 | Supernovae |
| | Swift SNe Ia[26] | 6 | Variable | 117 | Supernovae |
| Tabular | Gaia [59] | - | - | 220M | Stars |
| | PROVABGS [65] | - | - | 221K | Galaxy |
| | Galaxy10 DECaLS [147, 92] | - | - | 15K | Galaxy |

**Optical and Infrared Imaging**   Astronomical images are obtained via cameras installed at the focal plane of both space and ground-based telescopes, that capture photon emission from distant astronomical sources. An observed image $I$ can be described as: $I = S * \Pi + n$ where $S$ is the intrinsic source emission on the sky, $n$ is the measurement noise, $*$ is the convolution operation, and $\Pi$ is the instrumental response of the telescope, also known as the Point Spread Function (PSF). The PSF is at minimum characterized by its full width half maximum (fwhm), which sets the spatial resolution of the image. In addition, these images are typically acquired in a number of specific broad wavelength ranges, referred to as bands or channels. Knowledge of the bands, PSF, noise properties, and pixel scale, are generally required to fully characterize an observation and take into account the specificities of a particular instrument or observatory. Compared to natural images, astronomical images are typically noisier, and exhibit a large dynamic range, generally spanning several orders of magnitude between the bright objects and fainter sources.

**Spectroscopy**   Similarly to imaging data, light from distant objects is gathered and focused first by the telescope, but instead of being directly imaged with a camera, the light from cosmic objects is dispersed into various wavelengths or energies using a spectrograph. A spectrum is therefore a 1-d signal representing the decomposition of the light from an object as function of wavelength or energy of the photons. Similarly to images, a measured spectrum $y$ is the result of the following observational process: $y = L * x + n$ where $L$ is the instrument's impulse response (also known as Line Spread Function; LSF) and $n$ is a measurement noise. Contrary to images, where the absolute position of each pixel on the sky is not directly needed to interpret the signal, for spectroscopy knowledge of absolute wavelength corresponding to each pixel is crucial, which is why each spectrum not only provides a array of measured flux $y$ but also the corresponding wavelength array $\lambda$.

**Hyperspectral imaging**   Hyperspectral imaging is a data modality that combines imaging and spectroscopy to produce spatially resolved spectra in a two-dimensional region. It uses an Integral Field Unit (IFU) to produce a three-dimensional "datacube" consisting of 2d array of spatial pixel elements, i.e "spaxels", each containing a spectrum in a specific wavelength

Table 2: Description of standardized fields and metadata provided for the main modalities. These fields represent necessary and near-sufficient information to allow for the consistent interpretation of observations from multiple surveys or instruments.

| Modality | Field | Description |
|---|---|---|
| Images | flux | Array of flux measurements of the image |
| | ivar | Inverse variance of noise in the image |
| | band | Key indicating the wavelength range of the image |
| | psf_fwhm | Size of the instrumental response (Point Spread Function) |
| | scale | Scale of pixels on the sky |
| Spectra | flux | Measured flux as a function of wavelength |
| | ivar | Inverse variance of noise on measured flux |
| | lsf_sigma | Size of the instrumental response (Line Spread Function) |
| | lambda | Wavelength of each flux measurement |
| Time Series | flux | Measurements of flux as a function of time |
| | flux_err | Uncertainty on flux measurement |
| | band | Key indicating the wavelength range of the measurement |
| | time | Time of observation |

range. IFUs provide a wealth of information. For example, by analyzing the spectrum within each spaxel, one can produce "images" of spectral features and derived parameters, such as emission/absorption lines and the velocities of gas and stars in the galaxy. By aggregating spectra together within specific spatial regions, one can compare the properties (such as star forming rates) of different physical regions of the galaxy. By extension of both images and spectra, hyperspectral images are accompanied by an instrumental response in the form of both Point Spread Function and Line Spread Function information.

**Time-Series**   Time series data, often called light curves (brightness over time), are commonly examined when astronomical objects or events vary in brightness in time. These include relatively short-lived, explosive transients such as supernovae (exploding stars) and tidal disruption events, long-lived sources that show stochastic variation over time such as active galactic nuclei and quasars, and (quasi-)periodic sources such as pulsating stars and exoplanets. These time series are a sequence of flux (i.e. intensity) measurements. Because of their unique properties, astronomical time-series pose a challenge to traditional ML architectures. For one, most light curves are sparsely and irregularly sampled in both time and wavelength due to the unique survey strategy of each telescope. Flux measurements are also often plagued with large uncertainties, especially those from faint sources. Finally, data from different telescopes are highly heterogeneous and thus difficult to combine (e.g., to build a larger training set for ML applications). Together, these factors make our time-series data products a compelling and challenging dataset for ML researchers.

**Tabular data**   The MULTIMODAL UNIVERSE also aggregates large catalogs of tabular data from different sources, corresponding either to particular measurements (e.g. measured flux for a particular object), or the result of the processing of these observations by an analysis pipeline (e.g. calculations of galaxy properties or reporting of galaxy morphology)[1].

## 4   Examples of Machine Learning Tasks on Different Modalities

The data accumulated in the MULTIMODAL UNIVERSE covers a wide range of scientific uses-cases, spanning sub-fields of astrophysics from stellar physics to galaxy formation. We include in this section a few conventional examples of ML tasks in astrophysics as a way to illustrate how to interact with the MULTIMODAL UNIVERSE dataset and to provide illustrations of scientific uses cases. We note that these examples are far from exhaustive.

---

[1]We do not enforce standard definitions for tabular data and instead refer the user to the original survey description.

| Pretraining | Model | Top-1 Accuracy |
|---|---|---|
| No pretraining | EfficientNetB0 | **80.9** ±0.1 % |
| | ConvNext-nano [144] | 75.6 ±1.8 % |
| | ResNet18 | 73.9 ±0.9 % |
| | DenseNet121 | 73.5 ±2.4 % |
| Galazy Zoo | ConvNext-nano [144] | **89.3** ±0.1 % |
| ImageNet-12k | ConvNext-nano [144] | 83.9 ±0.3 % |

Table 3: Top-1 Accuracy on the Galaxy10 DECaLS morphology classification dataset.

## 4.1 Characterisation of Galaxies

**Morphology Classification**  Galaxy shape or morphology is a first-order tracer of the history of a galaxy. As such, classifying images of galaxies based on their apparent morphologies and visual features is a common task. Labeling these features is usually done by visual inspection from experts or using citizen science, such as [10, 145]. Here, we use the Galaxy10 DECaLS catalog of human-labeled annotations spanning ten broad morphological classes paired with the MULTIMODAL UNIVERSE's RGB-converted images from the Legacy Surveys and report the top-1 accuracy of supervised baselines on the held-out test set in Table 3. We include state of the art results from the literature from [144] as a point of reference and compare different common architectures for this task (EfficientNetB0[136], ResNet18[66], and DenseNet121[75]).

**Physical Property Inference**  Astrophysicists often seek to predict fundamental properties of galaxies from observational data. Accurate predictions of these properties provide key insights into galaxy formation and evolution. Here, we consider the following properties:

- Redshift ($Z_{HP}$): The extent to which the light from a galaxy has been stretched by the expansion of the universe, helping to determine the galaxy's distance from earth.
- Stellar Mass ($\log M_*$): The total mass of stars in a galaxy in units of solar masses.
- Age ($t_{age,MW}$): The age of stars in a galaxy, weighted by galaxy stellar mass.
- Specific Star-Formation Rate ($sSFR$): The rate at which stars in the galaxy are formed normalized by the galaxy stellar mass.
- Metallicity ($Z_{MW}$): The abundance of elements heavier than hydrogen and helium in the galaxy, typically weighted by the galaxy mass.

To implement this task, we use the MULTIMODAL UNIVERSE's built-in cross-matching functionality to combine the PROVABGS catalog of physical properties, with both the Legacy Survey North and South imaging sample and the DESI spectroscopic sample. We report the $R^2$ value on the held-out test set in Table 4.

Table 4: Model $R^2$ performance comparison for predicting galaxy properties from different observational data modalities.

| Modality | Source Survey | Model | $Z_{HP}$ | $\log M_*$ | $Z_{MW}$ | $t_{age,MW}$ | $sSFR$ |
|---|---|---|---|---|---|---|---|
| Image | Legacy Surveys | ResNet18 | 0.771 | 0.725 | 0.381 | 0.210 | 0.405 |
| | | DenseNet121 | 0.774 | 0.734 | 0.414 | 0.267 | 0.446 |
| | | EfficientNetB0 | 0.697 | 0.645 | 0.395 | 0.260 | 0.421 |
| Spectrum | DESI | Conv+Att [104] | 0.982 | 0.871 | 0.659 | 0.488 | 0.679 |
| Photometry | PROVABGS | MLP | 0.696 | 0.681 | 0.383 | 0.308 | 0.343 |

## 4.2 Characterisation of Transient events

**Candidate Identification**  Given their short-lived nature, early identification of transients is of great value to allow for further observations which study their evolution. The publicly-available BTSbot model [119, 120] - a multimodal convolutional neural network - was developed to automate identification of new astrophysical transient candidates in series of images and associated metadata. We report the performance of BTSbot on the binary classification of real astrophysical transients and bogus or uninteresting detections using a dataset from the Zwicky Transient Facility [ZTF; 15, 16] Bright Transient Survey [BTS; 56, 114].

Table 5: Results for astronomical time-series tasks. The performance of each model on the associated metric for the task is shown in the `Performance` column. Avocado is a Random Forest architecture while Connect Later is a transformer architecture.

| Source Survey | Task | Metric | Model | Performance |
|---|---|---|---|---|
| BTS | Transient Candidate Identification | AUC | BTSbot [119, 120] | 0.985 |
| YSE | SN Ia classification | AUC | Random forest | 0.90 |
| PLAsTiCC | 14-way classification | Accuracy | Avocado [20] | 77.4 |
| | 14-way classification | Accuracy | Connect Later [117] | 79.9 |
| | Redshift estimation | RMSE | Connect Later [117] | 0.247 |

**Light-curve Classification and Regression**  Classifying astrophysical sources can be challenging due to the sparsity of time-series data, limited filter coverage, or simply similarity between time-series of different object types. Spectroscopic data are required for unambiguous classification, but these follow-up observations will be prohibitively expensive for large surveys. Accurate photometric classification of astrophysical transients is thus highly desirable. This is a supervised learning problem, and has been attempted with both simulated [e.g. 83, 101, 107, 116, 71] and real training data [e.g. 143, 46, 30, 31, 91]. We provide baseline photometric classification results on a real dataset from the Young Supernova Experiment (YSE) as well as a simulated dataset from the Photometric LSST Astronomical Classification Challenge (PLAsTiCC). While we report state-of-the-art results from the literature on PLAsTiCC, YSE represents a specific challenge due to its limited size, and as such specifically provides an implementation of a Random Forest classifier, which is a conventional approach for handling limited dataset sizes. Details of these methods are included in Appendix B. Finally, as a related task, we also include an example of estimating redshift from light curves on the PLAsTiCC dataset. All metric performances are stated in Table 5.

## 4.3 Bridging the Modality Gap: Contrastive Image-Spectrum Pretraining

As a concrete example of the types of models that can be created thanks to the access to a large repository of multimodal data, we present a reproduction of the recent `AstroCLIP` method [112]. This variant of Contrastive Language-Image Pretraining (CLIP) [118] aims to build in a self-supervised representations of multimodal data by contrastive alignment between image and spectra modalities. `AstroCLIP` embeddings have been shown to perform in-line with supervised baselines on tasks like redshift prediction, galaxy property prediction, and morphology classification while also allowing for powerful in-modality and cross-modal retrieval. We demonstrate how MULTIMODAL UNIVERSE makes it possible to easily build such models by generating a cross-matched dataset of images from the Legacy Survey sample and optical spectra from the DESI sample. From this dataset we can train by contrastive learning image and spectra embeddings. We show in Table 6 k-Nearest Neighbour zero-shot regression results for redshift and galaxy property prediction, and show better performance compared to the supervised results reported in subsection 4.1.

Table 6: $R^2$ performance of zero-shot prediction of galaxy properties from image and spectrum `AstroCLIP` embeddings following the strategy described in [112].

| Modality | Source Survey | $Z_{HP}$ | $\log M_*$ | $Z_{MW}$ | $t_{age,MW}$ | $sSFR$ |
|---|---|---|---|---|---|---|
| Image | Legacy Surveys | 0.801 | 0.737 | 0.432 | 0.240 | 0.435 |
| Spectrum | DESI | 0.986 | 0.879 | 0.584 | 0.441 | 0.643 |

## 5  Discussion

**Paving the road towards Foundational Research in Scientific Machine Learning** Beyond its multimodal and large-scale nature, the MULTIMODAL UNIVERSE also serves as a valuable testbed for addressing some of the practical challenges encountered in the use of machine learning within application domains such as astrophysics and other scientific disciplines. Although machine learning has been applied in astronomy since at least the 1990s, and been extensively adopted in astrophysics research over the past decade, most

applications are tailored to specific datasets, requiring domain expertise, with models often being trained from scratch for similar purposes. Consequently, machine learning applications face issues such as distribution shifts [47, 155], uncertainty quantification [106], and the proper calibration of predictive models [78]. The Multimodal Universe is designed to facilitate the development of next-generation machine learning models in astrophysics, enabling research into each of these critical facets of applied machine learning in astronomy. We believe that the complexity, multimodal nature, scale, and the culture of publicly released data rooted in the astronomical community will enable the machine learning community at large to tackle some of the key challenges in large-scale machine learning development.

**Limitations**   While the Multimodal Universe includes a significant collection of multimodal scientific data, a notable omission is the lack of associated textual data. Providing image-text or observation-text pairs is not currently the norm for astronomical observations. Therefore, despite promising work on astronomy fine-tuned LLMs [109, 139], LLM evaluation [152], observation proposal based CLIP models [105], and NLP for improved language in astronomy [22, 23], the Multimodal Universe does not currently contain any text-based captions or corpora. An additional limitation is that the Multimodal Universe does not guarantee fully cross-matched samples (i.e. all modalities for each source). This is a consequence of observing strategies and science targets of the respective surveys. However, we note that the current outlook of astronomy surveys is towards 'all sky surveys'. As such, we expect that any future version of the Multimodal Universe will necessarily contain more cross-matched samples than are currently available.

# 6   Availability and Maintenance

The dataset is hosted in full at the Flatiron Institute and accessible both through HTTPS[2] or through GLOBUS[3] while associated resources and all source code necessary to reproduce the dataset compilation are hosted on GitHub. In addition, a preview of each dataset is hosted on the Hugging Face Hub in parquet format to allow for online streaming and easy prototyping. To ensure the continued evolution and relevance of Multimodal Universe, the project is organized as an open and collaborative effort. Contributions from researchers are actively encouraged, with a well-defined process for submitting new data, benchmarks, and improvements via GitHub. A dedicated team of maintainers oversees the integration of these contributions, ensuring that the dataset remains up-to-date and continues to meet the needs of the scientific and machine learning communities. This framework will enable regular updates to incorporate new data from ongoing and future astronomical surveys, as well as periodic evaluations and improvements based on feedback from the research community. Additionally, our open collaboration hosts forums via GitHub and a dedicated slack to engage with both ML and astrophysics communities, and help the Multimodal Universe adapt and evolve to the respective dynamic research environments.

## Acknowledgments and Disclosure of Funding

The authors would like to acknowledge the Center for Computational Astrophysics at the Flatiron Institute for hospitality while a portion of this work was carried out. In addition, the data used in this work are hosted on equipment supported by the Scientific Computing Core at the Flatiron Institute, a division of the Simons Foundation. Polymathic AI and MB gratefully acknowledge the support provided by Schmidt Sciences, LLC. MG and AD are supported by the European Union's Horizon 2020 research and innovation programme under ERC Grant Agreement No. 101002652 and Marie Skłodowska-Curie Grant Agreement No. 873089. BMB is supported by the Cambridge Centre for Doctoral Training in Data-Intensive Science funded by the UK Science and Technology Facilities Council (STFC) and a G-Research early career researchers grant used for equipment. EEH is supported by a Gates Cambridge Scholarship (#OPP1144). LP is supported by the NSF Graduate Research Fellowship Program. MHC acknowledges financial support from the State Research Agency

---

[2] https://users.flatironinstitute.org/~polymathic/data/MultimodalUniverse/
[3] https://app.globus.org/file-manager?origin_id=57136152-fc1d-418e-b74e-75ca52bddd21

(AEIMCINN) of the Spanish Ministry of Science and Innovation under the grants "Galaxy Evolution with Artificial Intelligence" with reference PGC2018-100852-A-I00 and "BASALT" with reference PID2021-126838NB-I00.

For data specific acknowledgments see each individual dataset in Appendix A.

## Contributors

The contribution categories listed below are broadly indicative of the ways in which authors contributed to the project, although we note these categories are not exhaustive.

**Coordination Team:** Micah Bowles, Marc Huertas-Company, Francois Lanusse, Liam H. Parker, Helen Qu, Michael J. Smith.

**Datasets:** Erini Angeloudi, Jeroen Audenaert, Micah Bowles, Benjamin M. Boyd, David Chemaly, Brian Cherinka, Aaron Do, Matthew Grayling, Erin E. Hayes, Tom Hehir, Marc Huertas-Company, Kartheik G. Iyer, Maja Jablonska, Francois Lanusse, Henry W. Leung, Kaisey Mandel, Juan Rafael Martínez-Galarza, Peter Melchior, Liam H. Parker, Helen Qu, Jeff Shen, Michael J. Smith, Connor Stone, Mike Walmsley, John F. Wu.

**Baselines:** Eirini Angeloudi, Jeroen Audenaert, Micah Bowles, Benjamin M. Boyd, David Chemaly, Aaron Do, Matthew Grayling, Erin Hayes, Tom Hehir, Marc Huertas-Company, Kaisey Mandel, Liam H. Parker, Helen Qu, Michael J. Smith.

**Overall/Infrastructure:** Micah Bowles, Ioana Ciucă, Miles Cranmer, Shirley Ho, Marc Huertas-Company, Francois Lanusse, Lucas Meyer, Liam H. Parker, Michael J. Smith, Mike Walmsley.

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

# Appendix Table of Contents

## A   Multimodal Universe Datasheet

### A.1  Motivation

The MULTIMODAL UNIVERSE dataset was compiled to enable the development of large-scale and multimodal scientific machine learning models and brings together hundreds of millions of astronomical observations, constituting over 70TB of data that includes multi-channel and hyperspectral images, spectra, multivariate time series, and scientific measurements from the field of astronomy. It further addresses a critical gap in the availability of standardized and machine learning-ready datasets in astronomy. Despite the data-rich nature of the field, existing datasets are often fragmented and tailored for traditional analysis rather than machine learning workflows. In addition, MULTIMODAL UNIVERSE puts significant emphasis on the inclusion and standardization of the metadata necessary to interpret these measurements, thus supporting the development of metadata-aware machine learning models which are necessary for the proper interpretation of scientific data. The data collected in MULTIMODAL UNIVERSE is gathered from a multitude of astronomical surveys and instruments, and is organized into subsets corresponding to one particular parent survey. The data is further organized as to allow the cross-matching of its different components in order to build pairs of different modalities for the same objects.

### A.2  Distribution

We openly distribute the MULTIMODAL UNIVERSE dataset under the Creative Commons Attribution (CC BY) 4.0 license, noting however that when using specific subsets, the license and conditions of utilisation should be respected. Any work using the MULTIMODAL UNIVERSE should therefore cite the main paper and appropriately cite or acknowledge the individual dataset sources, as described in this appendix. The Multimodal Universe package[4] offers citation tools to automatically retrieve the citations and acknowledgements of the various datasets.

The full dataset content in native HDF5 format is hosted at the Flatiron Institute. The data described in this paper is `v1` (version 1), with future versions expected to be hosted through the same system, and are available either through HTTPS or through GLOBUS[5] at the following links:

- https://users.flatironinstitute.org/~polymathic/data/
  MultimodalUniverse

---

[4] https://github.com/MultimodalUniverse/MultimodalUniverse/
[5] https://www.globus.org/

- 

GLOBUS is much preferable when downloading large amounts of data, or a large number of files. Local download of the full data in its native HDF5 format is necessary for using the provided cross-matching utilities.

In addition, for convenience, fast streaming access, and ease of prototyping, a copy of the dataset in parquet format is maintained on the Hugging Face Hub, as individual datasets under the MULTIMODAL UNIVERSE organization:

- https://huggingface.co/MultimodalUniverse

Finally, the source code used to compile all datasets, along with documentation and examples is hosted on the project's GitHub repository: https://github.com/MultimodalUniverse/MultimodalUniverse

### A.3 Image Datasets

### A.3.1 Legacy Surveys DR10: `legacysurvey` ☁️ 🐙 🤗 🥐

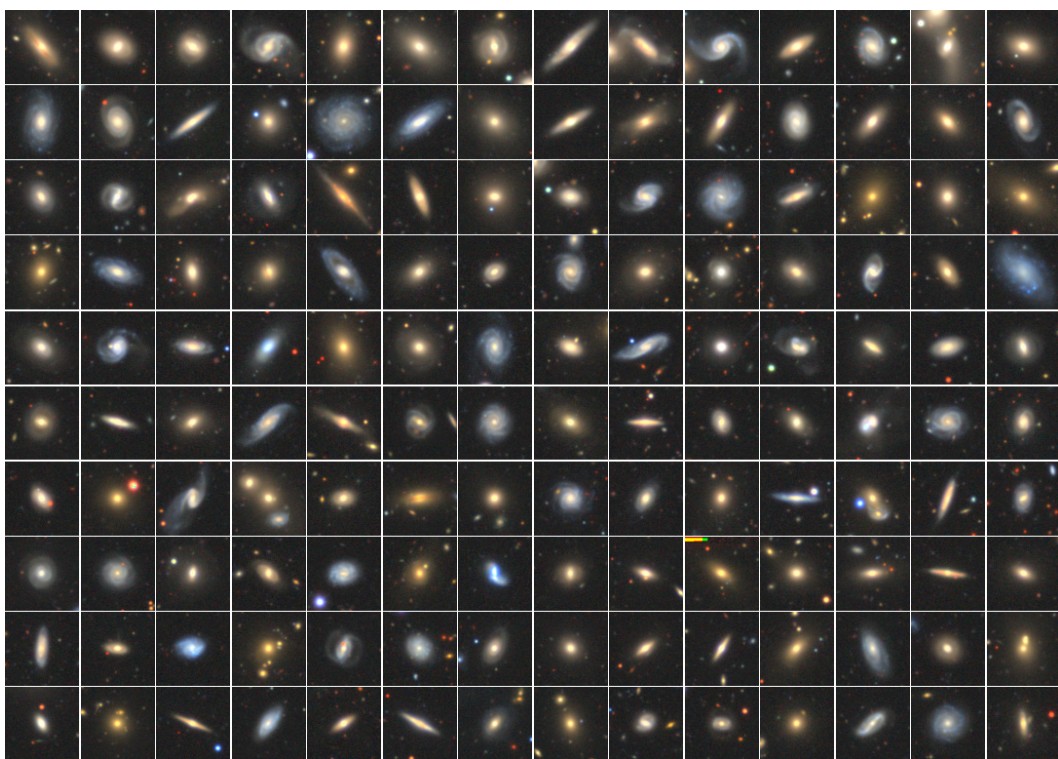

Figure 3: Illustration of grz images of galaxies from the Legacy Survey DR10 sample.

**Description**    The DESI Legacy Imaging Surveys (hereafter Legacy Surveys) [44] is originally a combination of dedicated large imaging surveys conducted with three different telescopes (the Blanco telescope at the Cerro Tololo Inter-American Observatory; the Mayall Telescope at the Kitt Peak National Observatory; and the University of Arizona Steward Observatory 2.3m (90inch) Bart Bok Telescope at Kitt Peak National Observatory), and covering over 14,000 square degrees in three optical bands (g,r,z). With its most recent data release, DR10, the Legacy Surveys were further extended in the southern hemisphere with additional observations in four bands (g,r,i,z) from the Dark Energy Camera (DECam), pushing the total survey area to almost 20,000 square degrees (i.e. half of the sky). Beyond the large joint footprint that the Legacy Surveys represent, their main characteristic is that they are

all analysed self-consistently with the same processing and inference pipeline, providing a large and uniform dataset over a large fraction of the sky. We specifically base this dataset on the 4-band imaging of the southern sky part of Data Release 10, and representing 15,000 square degrees.

**Composition**  The dataset we compile consists of 124M postage stamps of size 160x160 pixels with pixel size of 0.262 arcsec in the (g,r,i,z) bands, combined with a set of measurements from the Legacy Survey analysis pipeline. The sample was selected based on the following criteria:

- Extended objects: TYPE != 'PSF'

- Magnitude cut: mag_z < 21

- Full Color: observations in all 4 bands

- Quality cuts: all objects with the following flags are excluded[6]: BRIGHT, SATUR_G, SATUR_R, SATUR_Z ALLMASK_G, ALLMASK_R, ALLMASK_Z, MEDIUM, SATUR_I, ALLMASK_I

**Collection Process**  We first download from the Legacy Survey data archive[7] all coadded bricks from the DR10 southern region which possess observations in all 4 bands. Each of these bricks is an image of size 3600x3600 pixels, from which we extract 160x160 cutouts centered on the coordinates of detected objects that pass the selection cuts mentioned above. We further associate each of these cutouts with the corresponding entry in the Legacy Survey sweep catalogs which contain the most commonly used fields provided by the analysis pipeline.

**Preprocessing / cleaning**  No further processing is applied beyond the selection cuts reported above. This means in particular that cutouts may contain problematic regions (e.g. saturated stars) around the central object. These regions can be identified by the user through the `mask` field included with each cutout and that mark unreliable pixels in the image. The motivation for not including more aggressive cleaning is that defects are ubiquitous in astronomical images, and part of the motivation for this dataset is to build machine learning models that are robust to such defects.

**Uses**  It is the first time that this specific collection of data is assembled, however similar datasets based on Legacy Survey images have been used for a range of machine learning applications, such as identifying strong gravitational lenses [133, 76], or classifying galaxy morphologies [146].

**Acknowledgements and Terms of Use**  When used in scientific publications, the formal acknowledgement described at this URL should be used: https://www.legacysurvey.org/acknowledgment/#toc-entry-2

### A.3.2  Legacy Surveys North: `ssl_legacysurvey` ☁ 🞇 🤗 🥐

**Description**  This dataset is a collection of galaxy postage stamps extracted from the northern sky of the Legacy Surveys Data Release 9 in [134] in the context of building one of the first examples of self-supervised learning in astrophysics (hence the name `ssl_legacysurvey`). The northern component of DR9 contains (g,r,z) images from the Beijing-Arizona Sky Survey (BASS), and the Mayall z-band Legacy Survey (MzLS) which self consistently analysed through the Legacy Surveys pipeline. This dataset is complementary to subsubsection A.3.1 by providing a large set of imaging data in the northern part of the sky.

**Composition**  The dataset consists of 15M galaxies, each imaged in three optical bands $(g, r, z)$ at a pixel scale of 0.262 arcsec. The images are provided in $152 \times 152$ cut-outs. Each galaxy image is also associated with a set of measurement from the Legacy Survey catalog.

---

[6]Description available at: https://www.legacysurvey.org/dr10/bitmasks/
[7]https://www.legacysurvey.org/dr10/files/

The sample was selected based on the following criteria:

- Extended objects: TYPE != 'PSF'
- Magnitude cut: 20 < mag_z <21
- Quality cuts[8]: all objects with the following flags are excluded: BRIGHT, ALL-MASK_G, ALLMASK_R, ALLMASK_Z, MEDIUM

**Collection Process**   The original collection process used in [134] for this dataset is based on extracted cutouts from coadded images for each galaxy in the selected sample using the `imagine`[9] library. This processing was performed at the National Energy Research Scientific Computing Center (NERSC) using the local copy of the Legacy Survey data hosted at the facility. The collected image cutouts along with relevant catalog information was then packaged as a set of HDF5 files and made publicly available through GLOBUS [10]. The content of our dataset is directly and exactly extracted from the northern component of this dataset.

**Preprocessing/Cleaning**   No further processing is applied to the dataset compiled by [134].

**Uses**   This specific dataset was used to build the self-supervised representation learning model introduced in [134] and used to identify strong gravitational lensing in [132]. More recently, this dataset was used to build the image-spectrum contrastive learning model presented in [112].

**Acknowledgements and Terms of Use**   When using this dataset in scientific publications, the data compilation paper should be cited [132] and the formal acknowledgement described at this URL should be used: https://www.legacysurvey.org/acknowledgment/#toc-entry-2

### A.3.3   HSC: `hsc` ☁️ 🐙 🤗 🥐

**Description**   The Hyper Suprime-Cam Subaru Strategic Program (HSC-SSP) [5] provides a wide-field imaging survey using the Hyper Suprime-Cam on the Subaru Telescope. Covering approximately 1,200 square degrees of the sky in five broad-band filters (g, r, i, z, y), the survey boasts high-quality images with a median seeing of about 0.6 arcseconds in the i-band and depth reaching 26.4 magnitudes in the r-band, which leads to significantly better image quality than the Legacy Surveys. In this dataset, we specifically use data from the Deep/Ultra Deep fields of the third Public Data Release[PDR3, 4]. These fields represent a much smaller fraction of the sky (27 square degrees), but are much deeper (i.e. have less noise) than the wide survey.

**Composition**   The dataset is composed of 400k 160x160 cutouts in five bands (g,r,i,z,y) at a pixel scale of 0.168 arcsec. Each image is also associated with a set of measurements from the analysis pipeline. The sample was selected from the Deep/UltraDeep catalog based on the following criteria:

- Magnitude cut: mag_i < 22.5
- Full depth Full Color: At least 3 observations in all bands
- Quality cuts: Remove all objects affected by bright stars, objects intersecting edges, objects with saturated or interpolated pixels, and objects with unreliable cmodel magnitude.

These selection cuts lead to a sample of 400k objects (which may include stars in addition to galaxies).

---

[8]the meaning of the flags is described here: https://www.legacysurvey.org/dr9/bitmasks/

[9]https://github.com/legacysurvey/imagine

[10]https://github.com/georgestein/ssl-legacysurvey/tree/main?tab=readme-ov-file#data-access

**Collection Process**   We first download all coadded images (specifically `calexp` files from the `deepCoadd_result` folder) for the Deep/UltraDeep fields from the HSC PDR3 data archive[11]. Each of these images is of size 4200x4200, from which we extract cutouts centered on the coordinates of the objects in the sample selected above. We associate each of these cutouts with their corresponding entry in the object catalog.

**Preprocessing/cleaning**   No further processing is applied to the images beyond the selection cuts mentioned above. Similarly to previous imaging datasets, images can contain bad pixel or artifacts surrounding the central object, those can be identified through the `mask` field that identifies good pixels.

**Uses**   It is the first time that this specific dataset is compiled, it will have similar use cases as the other imaging datasets, i.e. machine learning methods for detecting rare objects, or classifying or characterizing galaxy properties.

**Acknowledgements and Terms of Use**   Data from the HSC SSP can be used without prior permission within the following scopes:

- Extent of free use stipulated by Japanese copyright law (private use, educational use, news reporting, etc.)
- Usage in academic research, education, and learning activities
- Usage by news organizations
- Usage in printed media, or in websites and social networks

The full description of restricted use-cases is provided here: https://www.nao.ac.jp/en/terms/copyright.html. In all cases, please explicitly include the credit "NAOJ / HSC Collaboration".

### A.3.4   BTSbot: `btsbot` ☁️ 🎧 🤗 🥐

**Description**   The Zwicky Transient Facility (ZTF) [15, 16] Bright Transient Survey (BTS) [56, 114] is the largest spectroscopic supernova survey ever constructed, surveying the entire nighttime Northern sky every 2-3 nights in two filters aiming to acquire spectra for a complete sample of bright, local transients. While identification of new transient sources has traditionally been led by human scanners assessing images, BTS have developed the BTSbot model and accompanying dataset [120] with the goal of automating transient identification. Each example in the BTSbot dataset corresponds to an 'alert', which will be generated automatically after observations and passed to the community to identify interesting objects and plan potential followup observations. The BTSbot model was trained specifically to identify sources expected to fit BTS's selection criteria.

**Composition**   The BTSbot dataset contains 409,107 transient alerts. Note that while each alert is treated separately, the same transient objects will feature multiple times in the data set. For example, real transients will be harder to identify in earlier alerts when they are fainter and data is noisier but will become more apparent in later alerts as they get brighter. Each transient alert comprises a set of 3 single-band images referred to as the 'science', 'reference', and 'difference' images. The science image shows the latest image of a potential transient event and its surrounding environment, for example the galaxy it is located in. The reference image is an archival image of the same area before the transient was present, while the difference image is the residual between the two which isolates the transient. We collect these views as different channels in a $3{\times}63{\times}63$ image in the same way that other multi-band imaging datasets organise images seen through different filters. The dataset is also multi-modal as each image triplet is accompanied by metadata containing extracted features of the transient source which are informative for classification.

---

[11]https://hsc-release.mtk.nao.ac.jp/archive/filetree/pdr3_dud/ (requires authentication)

**Collection Process**  A description of how the BTSbot dataset was compiled is given in the original paper [120]. We accessed the dataset at the authors' zenodo page[12].

**Preprocessing/Cleaning**  The BTSbot dataset is included within the MULTIMODAL UNIVERSE without any modifications (aside from reformatting).

**Uses**  The BTSbot dataset was used to develop the BTSbot model described in the original paper [120].

**Acknowledgements and Terms of Use**  The BTSbot dataset is distributed under the CC BY 4.0 license. Attribution is given to the authors of the BTSbot paper [120].

### A.3.5   JWST: `jwst` ☁ 🎝 🤗 🥐

**Description**  The JWST dataset contains NIRCam images from several of the first James Webb Space Telescope (JWST) deep field observations: CEERS [52], PRIMER [48], JADES [50] and NGDEEP [14].

**Composition**  Fixed size postage stamps (96 × 96 pixels) centered on the positions of sources in the photometric catalogs. Each deep survey is an independent instance of the JWST imaging survey and has a different wavelength coverage with 6 or 7 filters in the $\sim 0.9\mu m$ to $\sim 4.4\mu m$ range. The combined dataset contains of the order of 300K images. The list of available filters is: F090W ($0.9\mu m$), F115W ($1.15\mu m$), F150W ($1.5\mu m$), F200W ($2\mu m$), F277W ($2.7\mu m$), F356W ($3.56\mu m$), F444W ($4.4\mu m$).

**Collection Process**  For consistency, we use the public reductions and associated photometric catalogs from the Dawn JWST Archive[13]. Basic details of the NIRCam data reduction are presented in [141]. Images have been reduced with the grizli pipeline[14].

**Preprocessing/cleaning**  Where bands are missing, we store an array of zeros instead to simplify data loading.

**Uses**  Images from these deep JWST surveys have been used in a large number of scientific publications. However, the specific dataset compiled in this work has not been used in any prior publication. Some examples of machine learning works using JWST images are: [77, 142]

**Acknowledgements and Terms of Use**  All the data included in this dataset is of the public domain. The data products presented herein were retrieved from the Dawn JWST Archive (DJA). DJA is an initiative of the Cosmic Dawn Center (DAWN), which is funded by the Danish National Research Foundation under grant DNRF140.

## A.4   Time Series

### A.4.1   PLAsTiCC: `plasticc` ☁ 🎝 🤗 🥐

**Description**  The PLAsTiCC dataset is a collection of simulated light curves from the Photometric LSST Astronomical Time-Series Classification Challenge. This diverse dataset contains 14 types of astronomical time-varying objects, simulated using the expected instrument characteristics and survey strategy of the upcoming Legacy Survey of Space and Time [LSST 79] conducted at the Vera C. Rubin Observatory (Figure 4). The dataset includes two overall categories of time-series objects: *transients*, which are short-lived events such as supernovae, and *variable* sources, which are objects with fluctuating brightness such as pulsating stars. Specifically, the dataset includes the following transients: type Ia supernovae (SNIa), SNIax, SNIa-91bg, SNIbc, SNII, superluminous supernovae (SLSN), tidal disruption

---

[12]https://zenodo.org/records/10839691
[13]https://dawn-cph.github.io/dja/index.html
[14]https://github.com/gbrammer/grizli, https://zenodo.org/records/8370018

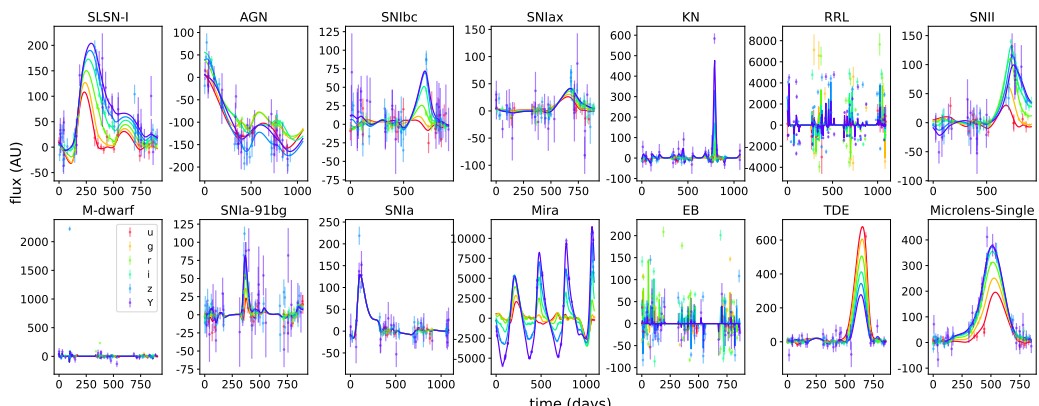

Figure 4: Examples of the 14 object types included in the PLAsTiCC dataset. Observations taken in different wavelength ranges, or *filters*, are shown in different colors. We overplot lines from Gaussian process interpolations of the data for visibility.

events (TDE), and single lens microlensing events ($\mu$Lens-Single); and the following variable objects: active galactic nuclei (AGN), Mira variables, eclipsing binary systems (EB), and RR Lyrae (RRL); details in Table 7.

Table 7: Breakdown of the PLAsTiCC train and test datasets by object type. The small training set size coupled with the class imbalances make this a challenging classification dataset.

| Object Type | Train Samples | Test Samples |
|---|---|---|
| SNIa | 2,313 | 1,659,831 |
| SNIa-91bg | 208 | 40,193 |
| SNIax | 183 | 63,664 |
| SNII | 1193 | 1,000,150 |
| SNIbc | 484 | 175,094 |
| SLSN-I | 175 | 35,782 |
| TDE | 495 | 13,555 |
| KN | 102 | 133 |
| AGN | 370 | 101,424 |
| RRL | 239 | 197,155 |
| M-dwarf | 981 | 93,494 |
| EB | 924 | 96,572 |
| Mira | 30 | 1,453 |
| $\mu$-Lens-Single | 151 | 1,303 |
| **Total** | **7,848** | **3,492,890** |

**Composition**  The dataset contains a collection of 3,492,890 time-series, providing in each case several fields: time of observation, flux, flux error, and filter in which the flux is measured. Observations in 6 filters are included (u,g,r,i,z,Y). In addition, each time series is associated with a class representing the type of object, and a redshift for that object.

**Collection Process**  The PLAsTiCC dataset is available on Zenodo at https://zenodo.org/records/2539456. We simply download and reformat the data to match the MULTI-MODAL UNIVERSE standard.

**Preprocessing/cleaning**  No further pre-processing is applied to the data.

**Uses**  The PLAsTiCC dataset has been widely used to develop and test supernova classification model [e.g. 116, 20].

**Acknowledgements and Terms of Use**  The dataset is released under the CC BY 4.0 License, it can be cited using the following DOI: 10.5281/zenodo.2539456.

### A.4.2 TESS: `tess` ☁️ 🐙 🤗 🥐

**Description**   The NASA Transiting Exoplanet Survey Satellite (TESS)[15] [121] is an all-sky photometric survey observing millions of sources to discover exoplanets and study variable stars. Stars are observed for a period of about 27 days at a time, before the satellite shifts to its next observing field (sector). Stars can be observed multiple times in different sectors over the course of the mission. The observations have a a cadence (sampling time) between 20 sec and 30 min, depending on the mission cycle and data product.

**Composition**   The TESS data set in the Multimodal Universe currently contains the light curves (brightness over time) for around 1,120,000 stars observed in sectors 56, 58, 60, 63, 64, 67 and 69, but the data can be downloaded for all available sectors by running the respective download scripts. The light curves consist of six fields: `object_id`, `time` (BTJD; Barycenter corrected TESS Julian Date), `flux` (electrons per second) and `flux_error` (electrons per second). In the future, we will also include the much larger set of light curves delivered by the MIT Quick-look Pipeline (QLP)[16] [73, 74, 87, 88], which contains up to a few million light curves per sector.

**Collection Process**   The TESS Science Processing Operations Center (SPOC) pipeline generates light curves [80] for up to 160,000 targets per sector [33] from the shorter cadence Target Pixel Files (TPFs) and the longer cadence Full-Frame Images (FFIs). The data sets were accessed through the Mikulski Archive for Space Telescopes (MAST) servers.

**Preprocessing/cleaning**   We use the Pre-search Data Conditioning (`PDC_SAP`) flux measurements, in which long term systematic trends have been removed using Co-trending Basis Vectors. We have further cleaned the light curves by applying a hard quality flag bitmask that removes data points with possible anomalies as described in the TESS Science Data Products Description Document and the TESS Data Release Notes.

**Uses**   Light curves delivered by TESS, or light curves with similar characteristics, such as those from the NASA Kepler mission [21, 85], have been used in a number of machine learning publications. Light curve classification (e.g., transiting exoplanet, pulsating star or eclipsing binary star) examples are [11, 72, 12, 137, 110, 37], while a stellar parameter estimation example is [111].

**Acknowledgements and Terms of Use**   The data collected by the TESS mission are publicly available from the Mikulski Archive for Space Telescopes (MAST). The light curves in this data set have been released under the CC BY 4.0 License. More detailed information on the data products can be found on MAST: https://archive.stsci.edu/hlsp/tess-spoc. Funding for the TESS mission is provided by NASA's Science Mission directorate.

### A.4.3 CfA Datasets: `cfa` ☁️ 🐙 🤗 🥐

**Description**   The Harvard-Smithsonian Center for Astrophysics (CfA) Supernova Group has published light curves of various classes of supernovae observed between 2000-2011.

**Composition**   The four datasets in the Multimodal Universe are as follows: CfA3 presents light curves of 185 Type Ia supernovae in the UBVRIr'i' bands in the standard photometric system, CfA4 provides light curves of and additional 94 Type Ia supernovae in the u'UBVr'i' bands in the standard photometric system, CfA-SECCSN presents light curves of 64 stripped-envelope core-collapse supernovae in the u'UBVRIr'i'JHK$_s$ bands. Lastly CfA-SNII has light curves of 60 Type II supernovae in the UBVRIu'r'i'JHKs bands.

We present an equivalent breakdown in Table 8.

Each dataset has a row for each supernova with two float fields for sky coordinates (ra, dec), two string fields (object_id and obj_type) describing each supernova's name and classification,

---

[15]https://tess.mit.edu/
[16]https://archive.stsci.edu/hlsp/qlp

Table 8: Breakdown of the CfA Samples.

| Source Survey | Shape | Number of samples | Main science |
|---|---|---|---|
| CfA 4 SNe Ia | Variable | 341 | Type I Supernovae |
| CfA 3 SNe Ia | Variable | 185 | Type I Supernovae |
| CfA Stripped-Core SNe | Variable | 249 | Stripped-envelope core-collapse Supernovae |
| CfA SNe II | Variable | 400 | Type II Supernovae |

and a lightcurve field with rows for each observation of that particular supernova. Each row has four subfields. The time subfield contains float values for the modified Julian date of observation (in units of days). The band subfield contains strings describing the band of the observation. The flux and flux_err fields are replaced by mag and mag_err fields, which present the original photometric measurements in the standard photometric system.

**Collection Process**   The compilation of each dataset is described in the corresponding papers [68, 69, 18, 70]. The datasets were accessed from the CfA Supernova Group's website https://lweb.cfa.harvard.edu/supernova/.

**Preprocessing/cleaning**   We have included supplemental data for each object, including RA, Dec, and spectroscopic classifications from the transient name server (TNS)[17], the International Astronomical Union's official repository for reporting astronomical transients.

**Uses**   These datasets were introduced in their respective papers, and have been used in many studies since. In particular, the Type Ia supernovae from CfA3 and CfA4 are included in the Pantheon [127] and Pantheon+ samples [128], the Joint Light-curve Analysis sample [17], the DESSN-3YR sample [25], the Union2 [9] and Union3 samples [123], and many more.

**Acknowledgements and Terms of Use**   This research has made use of the CfA Supernova Archive, which is funded in part by the National Science Foundation through grant AST 0907903. If you make use of CfA data, please include the same acknowledgement as was made here and cite the appropriate references for the data (see the 'Collection Process' paragraph above) in your publication. The CfA3 data is provided under the CC BY 4.0 license and the CfA-SNII data under the CC BY 3.0 license. Refer to the licensing policies of IOPscience[18] and AAS[19] for the CfA4 and CfA-SECCSN datasets.

### A.4.4   YSE: yse ☁ 🎃 🤗 🍜

**Description**   A collection of 1975 supernova light curves from the Young Supernova Experiment Data Release 1 (YSE DR1) [7]. An example use of this dataset would be for photometric classification.

**Composition**   We present the data as light curve objects containing time of observation, flux, flux error and band pass filter. Measurements are in $griz$ band pass filters. The supernova classification is included as metadata. Additional metadata includes information on coordinates, redshifts and host galaxy masses.

**Collection Process**   The data is presented in full as shown in the original paper [7] and is accessed through Zenodo[20].

**Acknowledgements and Terms of Use**   Original content from this work may be used under the terms of the Creative Commons Attribution 4.0 licence. Any further distribution of this work must maintain attribution to the author(s) and the title of the work, journal citation and DOI [7].

---

[17]https://www.wis-tns.org

[18]https://publishingsupport.iopscience.iop.org/creative-commons-licences/

[19]https://journals.aas.org/article-charges-and-copyright/

[20]https://zenodo.org/records/7317476

### A.4.5 PS1 SNe Ia: `ps1_sne_ia` ☁️ 🎧 🤗 🥐

**Description**  A collection of 369 spectroscopically confirmed type Ia supernova light curves from Pan-STARRS1 (PS1 SNe Ia) [127]. This dataset can be used for photometric redshift prediction or light curve inpainting.

**Composition**  We present the data as light curve objects containing time of observation, flux, flux error and band pass filter. Measurements are in *griz* band pass filters. The supernova classification is included as metadata. Additional metadata includes information on coordinates, redshifts and host galaxy masses.

**Collection Process**  The data is collected from the Pantheon+ compilation [128] that also performs a series of selection cuts. The data is accessed through the compilation's GitHub: https://github.com/PantheonPlusSH0ES/DataRelease/tree/main/Pantheon%2B_Data/1_DATA/photometry/Pantheon_PS1MD.

**Acknowledgements and Terms of Use**  Original content from the work [127] may be used under the terms of the Creative Commons Attribution 4.0 license. If you use this data, please cite the appropriate papers and acknowledge the dataset as stated on the Pan-STARRS webpage[21]. The Pan-STARRS1 Surveys (PS1) and the PS1 public science archive have been made possible through contributions by the Institute for Astronomy, the University of Hawaii, the Pan-STARRS Project Office, the Max-Planck Society and its participating institutes, the Max Planck Institute for Astronomy, Heidelberg and the Max Planck Institute for Extraterrestrial Physics, Garching, The Johns Hopkins University, Durham University, the University of Edinburgh, the Queen's University Belfast, the Harvard-Smithsonian Center for Astrophysics, the Las Cumbres Observatory Global Telescope Network Incorporated, the National Central University of Taiwan, the Space Telescope Science Institute, the National Aeronautics and Space Administration under Grant No. NNX08AR22G issued through the Planetary Science Division of the NASA Science Mission Directorate, the National Science Foundation Grant No. AST-1238877, the University of Maryland, Eotvos Lorand University (ELTE), the Los Alamos National Laboratory, and the Gordon and Betty Moore Foundation.

### A.4.6 DES Y3 SNe Ia: `des_y3_sne_ia` ☁️ 🎧 🤗 🥐

**Description**  A collection of 248 spectroscopically confirmed type Ia supernova light curves from the Dark Energy Survey Year 3 (DES Y3 SNe Ia) dataset [24]. This dataset can be used for photometric redshift prediction or light curve inpainting.

**Composition**  We present the data as light curve objects containing time of observation, flux, flux error and band pass filter. Measurements are in *griz* band pass filters. The supernova classification is included as metadata. Additional metadata includes information on coordinates, redshifts and host galaxy masses.

**Collection Process**  The data is collected from the Pantheon+ compilation [128] that also performs a series of selection cuts. The data is accessed through the compilation's GitHub: https://github.com/PantheonPlusSH0ES/DataRelease/tree/main/Pantheon%2B_Data/1_DATA/photometry/DES3YR_DES_COMBINED_TEXT.

**Acknowledgements and Terms of Use**  Original content from the work [24] may be used under the terms of the Creative Commons Attribution 4.0 license. If you use this data please cite [34, 24] and add the following acknowledgement to your work: This project used public archival data from the Dark Energy Survey (DES). Funding for the DES Projects has been provided by the U.S. Department of Energy, the U.S. National Science Foundation, the Ministry of Science and Education of Spain, the Science and Technology FacilitiesCouncil of the United Kingdom, the Higher Education Funding Council for England, the National Center for Supercomputing Applications at the University of Illinois at Urbana-Champaign, the Kavli Institute of Cosmological Physics at the University of Chicago, the

---

[21]https://outerspace.stsci.edu/display/PANSTARRS/

Center for Cosmology and Astro-Particle Physics at the Ohio State University, the Mitchell Institute for Fundamental Physics and Astronomy at Texas A&M University, Financiadora de Estudos e Projetos, Fundação Carlos Chagas Filho de Amparo à Pesquisa do Estado do Rio de Janeiro, Conselho Nacional de Desenvolvimento Científico e Tecnológico and the Ministério da Ciência, Tecnologia e Inovação, the Deutsche Forschungsgemeinschaft, and the Collaborating Institutions in the Dark Energy Survey. The Collaborating Institutions are Argonne National Laboratory, the University of California at Santa Cruz, the University of Cambridge, Centro de Investigaciones Energéticas, Medioambientales y Tecnológicas-Madrid, the University of Chicago, University College London, the DES-Brazil Consortium, the University of Edinburgh, the Eidgenössische Technische Hochschule (ETH) Zürich, Fermi National Accelerator Laboratory, the University of Illinois at Urbana-Champaign, the Institut de Ciències de l'Espai (IEEC/CSIC), the Institut de Física d'Altes Energies, Lawrence Berkeley National Laboratory, the Ludwig-Maximilians Universität München and the associated Excellence Cluster Universe, the University of Michigan, the National Optical Astronomy Observatory, the University of Nottingham, The Ohio State University, the OzDES Membership Consortium, the University of Pennsylvania, the University of Portsmouth, SLAC National Accelerator Laboratory, Stanford University, the University of Sussex, and Texas A&M University. Based in part on observations at Cerro Tololo Inter-American Observatory, National Optical Astronomy Observatory, which is operated by the Association of Universities for Research in Astronomy (AURA) under a cooperative agreement with the National Science Foundation.

### A.4.7   SNLS: `snls` ☁ ⌂ 🤗 🥐

**Description**   A collection of 239 spectroscopically confirmed type Ia supernova light curves from the Supernova Legacy Survey (SNLS) [63]. This dataset can be used for photometric redshift prediction or light curve inpainting.

**Composition**   We present the data as light curve objects containing time of observation, flux, flux error and band pass filter. Measurements are in $griz$ band pass filters. The supernova classification is included as metadata. Additional metadata includes information on coordinates, redshifts and host galaxy masses.

**Collection Process**   The data is collected from the Pantheon+ compilation [128] that also performs a series of selection cuts. The data is accessed through the compilation's GitHub: https://github.com/PantheonPlusSH0ES/DataRelease/tree/main/Pantheon%2B_Data/1_DATA/photometry/JLA2014_SNLS_DS17.

**Acknowledgements and Terms of Use**   Original content from the work [63] may be used under the terms of the GNU General Public License v3.0. If you use this data, please attribute the original data source [63] appropriately.

### A.4.8   Foundation: `foundation` ☁ ⌂ 🤗 🥐

**Description**   A collection of 180 spectroscopically confirmed type Ia supernova light curves from Foundation Data Release 3 [Foundation DR3 53, 81]. This dataset can be used for photometric redshift prediction or light curve inpainting.

**Composition**   We present the data as light curve objects containing time of observation, flux, flux error and band pass filter. Measurements are in $griz$ band pass filters. The supernova classification is included as metadata. Additional metadata includes information on coordinates, redshifts and host galaxy masses.

**Collection Process**   The data is collected from the Pantheon+ compilation [128] that also performs a series of selection cuts. The data is accessed through the compilation's GitHub: https://github.com/PantheonPlusSH0ES/DataRelease/tree/main/Pantheon%2B_Data/1_DATA/photometry/Foundation_DJ17.

### A.4.9   CSP: `csp` ☁ 🐙 🤗 🥐

**Description**   The third data release of the first stage of the Carnegie Supernova Project (CSP) contains light curves of 134 spectroscopically confirmed Type Ia supernovae observed between 2004-2009 [86].

**Composition**   Aside from the use of different bands (ugriBVYJH), the data fields are identical to those of the CfA datasets.

**Collection Process**   The compilation of the CSP data is described in the data release paper [86]. The dataset was accessed from the CSP website's data products page `https://csp.obs.carnegiescience.edu/data`.

**Preprocessing/cleaning**   No pre-processing has been applied.

**Uses**   The CSP dataset has been used in many studies including those mentioned in the CfA section, as well as several CSP studies [e.g. 32, 115, 55, 140].

### A.4.10   Swift SNe Ia: `swift_sne_ia` ☁ 🐙 🤗 🥐

**Description**   A collection of 117 spectroscopically confirmed type Ia supernova light curves from the Swift Type Ia Supernova (Swift SN Ia) dataset [27]. This dataset can be used for photometric redshift prediction or light curve inpainting.

**Composition**   We present the data as light curve objects containing time of observation, flux, flux error and band pass filter. Measurements are in $UV_{w1}$, $u$, $b$, $UV_{w2}$, $v$ and $UV_{m2}$ band pass filters. The supernova classification is included as metadata. Additional metadata includes information on coordinates, redshifts and host galaxy masses.

**Collection Process**   The data is collected from the Pantheon+ compilation [128] that also performs a series of selection cuts. The data is accessed through the compilation's GitHub: `https://github.com/PantheonPlusSH0ES/DataRelease/tree/main/Pantheon%2B_Data/1_DATA/photometry/SWIFT`.

## A.5   Spectra Datasets

### A.5.1   *Gaia* BP/RP: `gaia` ☁ 🐙 🤗 🥐

Refer to Subsection A.7.1.

### A.5.2   SDSS: `sdss` ☁ 🐙 🤗 🥐

**Description**   The Sloan Digital Sky Survey (SDSS; [153]) is a comprehensive astronomical survey that has mapped the sky using a dedicated 2.5-meter wide-angle optical telescope located at the Apache Point Observatory in New Mexico, USA. SDSS has collected detailed images and spectra of millions of celestial objects, covering a significant portion of the sky in multiple bands. In this dataset we collect data from all SDSS optical spectra including SDSS Legacy survey, SEGUE-1, SEGUE-2, BOSS, and eBOSS surveys [153, 49, 19].

**Composition**  The SDSS dataset includes spectra spanning a wavelength range from $3,800$ to $9,200$ for spectra acquired with the SDSS spectrograph and $3,650$-$10,400$ for data from the BOSS spectrograph. In addition, for various processing reasons, not all spectra have the same lengths. Each spectrum comes with associated metadata, including flux measurements, wavelength values, and inverse variance (`ivar`) for the flux. Additionally, pixel-level masks indicate potential quality issues. The dataset contains a total of roughly 4 million spectra.

We apply a minimal selection on this data by only imposing the following cuts:

- Avoid duplicated spectra for the same object: SPECPRIMARY = 1
- Only good plates: PLATEQUALITY = 'good'
- Only science targets (no sky, calibration, etc.): TARGETTYPE = 'science'

**Collection Process**  We use the publicly accessible data from the SDSS $18^{th}$ data release (SDSS DR18) [22], and systematically download all processed spectra files for spectra that pass the selection cut described above.

**Preprocessing/Cleaning**  No further cuts of processing steps are performed.

**Uses**  SDSS spectra have been used in numerous scientific publications. Notable examples involving machine learning include building data-driven representations of galaxy spectra and looking for outliers [104].

**Acknowledgements and Terms of Use**  All SDSS data released in public data releases is considered to be in the public domain. When using the data included in this compilation, the following acknowledgements should be included:

- **SDSS and SEGUE-1 samples**: Funding for the SDSS and SDSS-II has been provided by the Alfred P. Sloan Foundation, the Participating Institutions, the National Science Foundation, the U.S. Department of Energy, the National Aeronautics and Space Administration, the Japanese Monbukagakusho, the Max Planck Society, and the Higher Education Funding Council for England. The SDSS Web Site is http://www.sdss.org/.
  The SDSS is managed by the Astrophysical Research Consortium for the Participating Institutions. The Participating Institutions are the American Museum of Natural History, Astrophysical Institute Potsdam, University of Basel, University of Cambridge, Case Western Reserve University, University of Chicago, Drexel University, Fermilab, the Institute for Advanced Study, the Japan Participation Group, Johns Hopkins University, the Joint Institute for Nuclear Astrophysics, the Kavli Institute for Particle Astrophysics and Cosmology, the Korean Scientist Group, the Chinese Academy of Sciences (LAMOST), Los Alamos National Laboratory, the Max-Planck-Institute for Astronomy (MPIA), the Max-Planck-Institute for Astrophysics (MPA), New Mexico State University, Ohio State University, University of Pittsburgh, University of Portsmouth, Princeton University, the United States Naval Observatory, and the University of Washington.
- **BOSS and SEGUE-2 samples**: Funding for SDSS-III has been provided by the Alfred P. Sloan Foundation, the Participating Institutions, the National Science Foundation, and the U.S. Department of Energy Office of Science. The SDSS-III web site is http://www.sdss3.org/.
  SDSS-III is managed by the Astrophysical Research Consortium for the Participating Institutions of the SDSS-III Collaboration including the University of Arizona, the Brazilian Participation Group, Brookhaven National Laboratory, Carnegie Mellon University, University of Florida, the French Participation Group, the German Participation Group, Harvard University, the Instituto de Astrofisica de Canarias, the Michigan State/Notre Dame/JINA Participation Group, Johns Hopkins University, Lawrence Berkeley National Laboratory, Max Planck Institute for Astrophysics, Max Planck Institute for Extraterrestrial Physics, New Mexico State University, New

---

[22]https://www.sdss.org/dr18/data_access/bulk/

York University, Ohio State University, Pennsylvania State University, University of Portsmouth, Princeton University, the Spanish Participation Group, University of Tokyo, University of Utah, Vanderbilt University, University of Virginia, University of Washington, and Yale University.

- **eBOSS sample**: Funding for the Sloan Digital Sky Survey IV has been provided by the Alfred P. Sloan Foundation, the U.S. Department of Energy Office of Science, and the Participating Institutions. SDSS acknowledges support and resources from the Center for High-Performance Computing at the University of Utah. The SDSS web site is www.sdss4.org.

  SDSS is managed by the Astrophysical Research Consortium for the Participating Institutions of the SDSS Collaboration including the Brazilian Participation Group, the Carnegie Institution for Science, Carnegie Mellon University, Center for Astrophysics | Harvard & Smithsonian (CfA), the Chilean Participation Group, the French Participation Group, Instituto de Astrofísica de Canarias, The Johns Hopkins University, Kavli Institute for the Physics and Mathematics of the Universe (IPMU) / University of Tokyo, the Korean Participation Group, Lawrence Berkeley National Laboratory, Leibniz Institut für Astrophysik Potsdam (AIP), Max-Planck-Institut für Astronomie (MPIA Heidelberg), Max-Planck-Institut für Astrophysik (MPA Garching), Max-Planck-Institut für Extraterrestrische Physik (MPE), National Astronomical Observatories of China, New Mexico State University, New York University, University of Notre Dame, Observatório Nacional / MCTI, The Ohio State University, Pennsylvania State University, Shanghai Astronomical Observatory, United Kingdom Participation Group, Universidad Nacional Autónoma de México, University of Arizona, University of Colorado Boulder, University of Oxford, University of Portsmouth, University of Utah, University of Virginia, University of Washington, University of Wisconsin, Vanderbilt University, and Yale University.

### A.5.3   DESI: `desi` 🌨️ 🎏 🤗 🥐

**Description**   The Dark Energy Spectroscopic Instrument (DESI; [41]) is designed to measure the effect of dark energy on the expansion of the universe by mapping the distributions of galaxies in three dimensions. To do so, DESI uses a 4-meter telescope at Kitt Peak National Observatory to collect spectra from millions of galaxies and quasars over a five-year period. This experiment is still on-going and most of the data is not yet public. In this dataset, we specifically make use of the DESI Early Data Release (EDR) [40] which contains 1% survey of the final survey.

**Composition**   The DESI EDR sample includes spectra with a fixed wavelength range from $3,600$ to $9,800$ and 7081 pixels. Each spectrum is accompanied by flux measurements, wavelength values, and inverse variance (`ivar`) for the flux. To select the sample including in this dataset we only apply the following minimal set of selection cuts:

- Only use data from the one percent survey: SURVEY = 'sv3'

- Only use the primary spectrum for each object: SV_PRIMARY is true

- Only use targets (ignore sky and others): OBJTYPE = 'TGT'

- Only use fibers with good status: COADD_FIBERSTATUS = 0

This results in a set of approximately 1M spectra of galaxies, quasars, and stars.

**Collection Process**   We systematically download all spectra files for the SV3 sample through GLOBUS from the DESI EDR archive https://data.desi.lbl.gov/doc/releases/.

**Preprocessing/Cleaning**   We apply no further pre-processing beyond the set of cuts described above.

**Uses**  The DESI EDR being still very recent, few machine learning works have been developed on this dataset. Still, a few examples are the `AstroClip` multimodal representation learning model [112], and the detection of outliers with the `spender` spectrum auto-encoding model [96].

**Acknowledgements and Terms of Use**  The Dark Energy Spectroscopic Instrument (DESI) data are licensed under the Creative Commons Attribution 4.0 International License ("CC BY 4.0"). Users are free to share, copy, redistribute, adapt, transform and build upon the DESI data available through https://data.desi.lbl.gov/ for any purpose, including commercially.

When DESI data are used, the appropriate credit is required by using both the following reference [40] and acknowledgements text:

This research used data obtained with the Dark Energy Spectroscopic Instrument (DESI). DESI construction and operations is managed by the Lawrence Berkeley National Laboratory. This material is based upon work supported by the U.S. Department of Energy, Office of Science, Office of High-Energy Physics, under Contract No. DE–AC02–05CH11231, and by the National Energy Research Scientific Computing Center, a DOE Office of Science User Facility under the same contract. Additional support for DESI was provided by the U.S. National Science Foundation (NSF), Division of Astronomical Sciences under Contract No. AST-0950945 to the NSF's National Optical-Infrared Astronomy Research Laboratory; the Science and Technology Facilities Council of the United Kingdom; the Gordon and Betty Moore Foundation; the Heising-Simons Foundation; the French Alternative Energies and Atomic Energy Commission (CEA); the National Council of Science and Technology of Mexico (CONACYT); the Ministry of Science and Innovation of Spain (MICINN), and by the DESI Member Institutions: www.desi.lbl.gov/collaborating-institutions. The DESI collaboration is honored to be permitted to conduct scientific research on Iolkam Du'ag (Kitt Peak), a mountain with particular significance to the Tohono O'odham Nation. Any opinions, findings, and conclusions or recommendations expressed in this material are those of the author(s) and do not necessarily reflect the views of the U.S. National Science Foundation, the U.S. Department of Energy, or any of the listed funding agencies.

### A.5.4   APOGEE: apogee ☁️ 🐙 🤗 🥐

**Description**  Apache Point Observatory Galactic Evolution Experiment (APOGEE; [19, 2]) within the Sloan Digital Sky Survey (SDSS) is a high-resolution ($R \sim 22,000$), high signal-to-noise ( 100 per pixel typically) stellar spectroscopic survey with 2.5-m telescopes in both northern and southern hemisphere in the near infrared H-band wavelength region [102, 151].

**Composition**  Fixed size APOGEE stellar spectra ($1 \times 7514$) spanning a wavelength range of 1.5 to $1.7\mu m$. Each stellar spectrum comes with both flux and continuum normalized flux (i.e. flux with continuum removed), its associated $\lambda$ vector representing the corresponding wavelength of each pixel, as well the inverse variance (`ivar`) for both flux and continuum normalized flux and a pixel-level mask indicating potential quality issues. The dataset contains a total of roughly 700,000 stellar spectra. Surface temperature $T_{\mathrm{eff}}$, surface gravity $\log g$, overall metallicity $[M/H]$, overall alpha abundance $[\alpha/H]$ and radial velocity are derived using the APOGEE Stellar Parameter and Chemical Abundances Pipeline [61] from the APOGEE spectra.

**Collection Process**  We use the publicly accessible data from the APOGEE $17^{th}$ data release (APOGEE DR17) catalog [23].

**Preprocessing/cleaning**  The dataset only include APOGEE spectra that has signal-to-noise (SNR) greater than 30, where duplicated stars in the public data released are removed due to some stars being observed in multiple programs multiple times. Preprocessing of the raw data are done internally hence minimal processing are needed for the public data release which we are using.

---

[23]https://www.sdss4.org/dr17/irspec/

**Uses** The public APOGEE DR17 dataset has been used in a number of scientific publications in both stellar and Galactic astronomy, as well as in machine learning specific applications.

**Acknowledgements and Terms of Use** Funding for the Sloan Digital Sky Survey IV has been provided by the Alfred P. Sloan Foundation, the U.S. Department of Energy Office of Science, and the Participating Institutions. SDSS-IV acknowledges support and resources from the Center for High Performance Computing at the University of Utah. The SDSS website is www.sdss.org. All SDSS data released in our public data releases is considered in the public domain. If you use APOGEE data in your work, please cite and acknowledge it as above.

### A.5.5 GALAH: `galah` ☁️ 🔾 🤗 🥐

**Description** Galactic Archaeology with HERMES (GALAH; [39]) is an Australian-led survey aimed at obtaining high-quality one-dimensional stellar spectra for deriving stellar atmospheric parameters and elemental abundances.

**Composition** The dataset includes spectra and derived stellar parameters such as effective temperature and several elemental abundances. The publicly available Data Release 3 (DR3; [28]) contains spectra for 325,518 objects after applying recommended quality cuts.

**Collection Process** The GALAH survey utilizes the HERMES spectrograph on the Anglo-Australian Telescope to collect high-resolution spectra. The observations are processed and made publicly available through periodic data releases, with DR3 being the latest release used for this dataset. We download the data through https://www.galah-survey.org/dr3/the_spectra/

**Preprocessing/Cleaning** We apply recommended cuts on spectra quality[24] to ensure the reliability of the data, resulting in a dataset with high-quality spectra and accurate stellar parameters.

**Uses** The GALAH dataset is used in studies of stellar populations, Galactic archaeology, and the chemical evolution of the Milky Way.

**Acknowledgements and Terms of Use** The GALAH data is openly accessible, When using GALAH data, please cite [39, 28] and provide the following acknowledgements:

This work made use of the Third Data Release of the GALAH Survey (Buder et al. 2021). The GALAH Survey is based on data acquired through the Australian Astronomical Observatory, under programs: A/2013B/13 (The GALAH pilot survey); A/2014A/25, A/2015A/19, A2017A/18 (The GALAH survey phase 1); A2018A/18 (Open clusters with HERMES); A2019A/1 (Hierarchical star formation in Ori OB1); A2019A/15 (The GALAH survey phase 2); A/2015B/19, A/2016A/22, A/2016B/10, A/2017B/16, A/2018B/15 (The HERMES-TESS program); and A/2015A/3, A/2015B/1, A/2015B/19, A/2016A/22, A/2016B/12, A/2017A/14 (The HERMES K2-follow-up program). We acknowledge the traditional owners of the land on which the AAT stands, the Gamilaraay people, and pay our respects to elders past and present. This paper includes data that has been provided by AAO Data Central (datacentral.org.au).

### A.5.6 VIPERS: `vipers` ☁️ 🔾 🤗 🥐

**Description** The "VIMOS Public Extragalactic Redshift Survey" (VIPERS) [126] dataset contains measured optical spectra of galaxies in the $0.5 < z < 1.0$ range.

**Composition** Fixed size spectra ($1 \times 557$) spanning a wavelength range of 5514 Å to 9484 Å, taken at the center of each galaxy. Each spectrum comes with its associated $\lambda$ vector,

---

[24]https://www.galah-survey.org/dr3/using_the_data/

as well the inverse variance (`ivar`) and a mask of each flux measurement in the spectrum. The dataset contins a total of roughly 90,000 galaxies with red magnitude I(AB) brighter than 22.5 over an overall area of nearly 24 square degrees.

**Collection Process**   For consistency, we use the publicly accessible data from the second VIPERS public data release (PDR-2) catalog[25].

**Preprocessing/cleaning**   The VIPERS spectra flux is natively stored in units of erg cm$^{-2}$s$^{-1}$Å$^{-1}$, and correspondingly the noise is stored in erg cm$^{-2}$s$^{-1}$Å$^{-1}$ and the wavelength in angstroms (Å). To be consistent with the other spectra datasets, we normalize the spectra fluxes to $10^{-17}$erg cm$^{-2}$s$^{-1}$Å$^{-1}$. Correspondingly, we convert the spectra noise to inverse variance, also in units $10^{-17}$erg cm$^{-2}$s$^{-1}$Å$^{-1}$.

**Uses**   This dataset has been used in a number of scientific publications, as well as in machine learning specific applications, including source identification with SVMs [103] and galaxy classification [129] with unsupervised methods.

**Acknowledgements and Terms of Use**   This paper uses data from the VIMOS Public Extragalactic Redshift Survey (VIPERS). VIPERS has been performed using the ESO Very Large Telescope, under the "Large Programme" 182.A-0886. The participating institutions and funding agencies are listed at http://vipers.inaf.it.

### A.5.7   Chandra: `chandra` ☁ 🐙 🤗 🥐

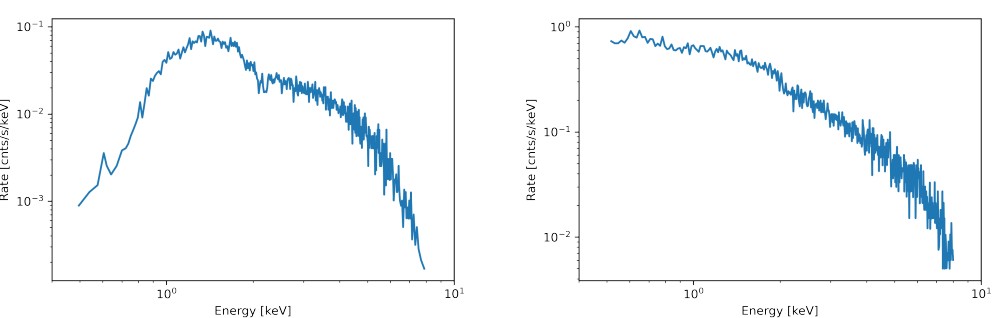

Figure 5: Examples of Chandra X-ray spectra included in MULTIMODAL UNIVERSE. Spectra of sources 2CXO J002029.1+591651 (left) and 2CXO J023437.8-084716 (right) are shown.

**Description**   The Chandra X-ray Observatory is one of NASA's flagship observatories, studying the X-ray emission from the most extreme environments in the Universe, including the surroundings of stellar-mass black holes and super massive black holes (SMBHs), supernova remnants, and violent coronal mass ejections from young stars [150]. The observatory records these observations using two instruments, the Advanced CCD Imaging Spectrometer (ACIS), and the High Resolution Camera (HRC), with CCD chips arranged to cover the telescope's field of view. The Chandra processing pipeline produces the individual X-ray photon recordings for each observation in the form of event files, which can be thought of as multivariate time series of the photon's energies and spatial location on the detector, with the energies covering the range between approximately 0.5 keV and 8 keV. MULTIMODAL UNIVERSE contains ACIS X-ray spectra of astrophysical sources, derived from the event files through the proper application of the instrument's spectral response.

X-ray spectra (photon energy versus count rate) from astrophysical sources detected by the Chandra X-ray Observatory in the energy range between 0.5keV and 8keV, binned at 10 counts per energy bin. The spectral files are downloaded from the Chandra Source Catalog

---

[25]http://www.vipers.inaf.it/rel-pdr2.html

(CSC) version 2.1 server. (`https://cxc.cfa.harvard.edu/csc/`). We provide all CSC spectra that have at least 40 total counts and a detection signal to noise ratio of 4 or larger.

**Composition**   Each entry consist of the source identifiers and coordinates, as well as arrays that contain the low and high limits of the energy bins, the center of the bin, the measured count rate per bin, and the error in that count rate. In addition, we provide features that characterize the observed source: the aperture photometry X-ray flux in physical units (ergs s$^{-1}$ cm$^{-2}$), the flux estimated using a blackbody fit to the spectrum, the flux significance, or signal-to-noise ratio, three hardness ratios which characterize the spectral shape, and measures of the flux variability as a function of time, both as a probability and as an index that quantifies the statistical significance of the variability.

**Collection Process**   The data was originally collected as part of the Chandra Source Catalog project, which detects and characterizes all sources serendipitously detected by Chandra, and produces both tables and data products for those sources, including spectral files, storing them as a database in a dedicated server hosted at the Smithsonian Astrophysical Observatory. For MULTIMODAL UNIVERSE, we have accessed these files using International Virtual Observatory (IVOA) interfaces and HTTP requests via command line interface.

**Preprocessing/cleaning**   The observatory records these observations using two instruments, the Advanced CCD Imaging Spectrometer (ACIS), and the High Resolution Camera (HRC), with CCD chips arranged to cover the telescope's field of view. The Chandra processing pipeline produces the individual X-ray photon recordings for each observation in the form of event files, which can be thought of as multivariate time series of the photon's energies and spatial location on the detector, with the energies covering the range between approximately 0.5 keV and 8 keV. MULTIMODAL UNIVERSE contains ACIS X-ray spectra of astrophysical sources, derived from the event files through the proper application of the instrument's spectral response, and processed using the Chandra Interactive Analysis of Observations (CIAO) software (`https://cxc.cfa.harvard.edu/ciao/`), in particular the Sherpa package.

**Uses**   Chandra studies the X-ray emission from the most extreme environments in the Universe, including the surroundings of stellar-mass black holes and super massive black holes (SMBHs), supernova remnants, and violent coronal mass ejections from young stars [150]. Many of the serendipitous sources compiled in the CSC lack and astrophysical classification. A similar dataset to the one presented here has been used to perform supervised machine learning classification of Chandra sources [67].

**Acknowledgements and Terms of Use**   This research has made use of data obtained from the Chandra Source Catalog, provided by the Chandra X-ray Center (CXC). The CSC (doi:10.25574/csc2) is publicly available. The current version is CSC 2.1. CXC provides up to date guidance[26] on acknowledging and citing their work, which we ask you do if you use Chandra data.

### A.6   HyperSpectral Imaging Datasets

### A.6.1   MaNGA `manga` ☁ ⭕ 🤗 🐘

**Description**   The SDSS-IV MaNGA survey[29] is a wide-field, optical, IFU survey of 10,000 nearby galaxies. MaNGA's goal is to understand the "life history" of present day galaxies from imprinted clues of their birth and assembly, through their ongoing growth via star formation and merging, to their death from quenching at late times.

The primary MaNGA data products are composed of 3-D calibrated data cubes produced by the DRP[89] and 2-D maps of derived quantities, such as emission line fluxes, gas and stellar kinematics, and stellar population properties, produced by the DAP[149] from those cubes. The 3-D data cubes are constructed from a few tens to a few thousands of individual spectra that have been combined onto a regular grid. The 2-D maps of derived quantities

---

[26]`https://cxc.cfa.harvard.edu/csc2.1/cite.html`

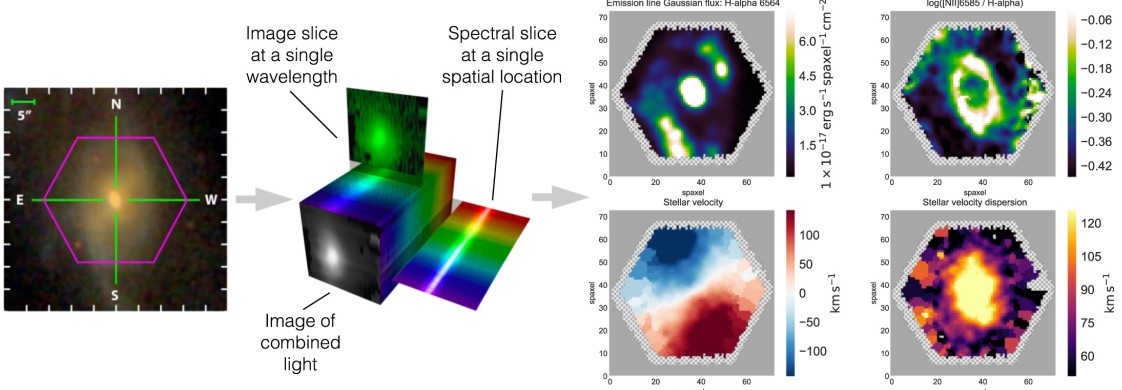

Figure 6: Left: gri image of MaNGA 1-596678 with the IFU field of view shown in purple. Middle: IFU observations produce three-dimensional data cubes, with one spectral and two spatial dimensions (credit: Stephen Todd and Douglas Pierce-Price; IFS Wiki http://ifs.wikidot.com). Right: spectral analysis of individual spaxels produces hundreds of two-dimensional maps for each galaxy spanning a wide range of physical properties. The four example maps shown for 1-596678 are flux (top left), log([N ii] 6585/) flux (top right), stellar velocity (bottom left), and stellar velocity dispersion corrected for instrumental broadening (bottom right). Original figure from the SDSS MaNGA Marvin paper,[35].

are constructed by analyzing individual or binned groups of spaxels and constructing maps of the quantities at the relevant on-sky location.

This dataset contains information pulled from the MaNGA DRP LOGCUBE[27] files, as well as the DAP "HYB10-MILESHC-MASTARSSP" MAPS[28] files.

**Composition**   Each entry in the dataset is a single MaNGA galaxy observation, identified by its unique id designation, the "plate-IFU", e.g. "8485-1901", with information aggregated from both the IFU data cubes and the derived analsyis maps, for that target. For each galaxy, the following features are provided:

- **object__id**: the MaNGA plate-IFU
- **ra**: the Right Ascension of the target
- **dec**: the Declination of the target
- **healpix**: the healpix id
- **z**: the target redshift
- **spaxel__size**: the size of the spaxel (spatial pixel)
- **spaxel__size__units**: the units of the spaxel size
- **spaxels** a list of spaxels, with properties
- **images**: a list of reconstructed images, with properties
- **maps**: a list of derived maps, with properties

Each **spaxel** item has the following features:

- **flux**: the flux spectrum array in the spaxel
- **ivar**: the inverse variance array in the spaxel
- **mask**: the pixel quality mask array in the spaxel
- **lsf**: the line-spread-function array in the spaxel

---

[27]https://www.sdss4.org/dr17/manga/manga-data/data-model/#CUBEFiles
[28]https://www.sdss4.org/dr17/manga/manga-data/data-model/#DataAnalysisPipelineOutput

- **lambda**: the wavelength array, same for all spaxels
- **x**: the x-array index spaxel coordinate
- **y**: the y-array index spaxel coordinate
- **spaxel_idx**: the 1d array index of the numpy.raveled spaxel
- **flux_units**: the physical units of the flux array
- **lambda_units**: the physical units of the wavelength array
- **skycoo_x**: the sky x offset in RA, in arcsec from galaxy center
- **skycoo_y**: the sky y offset in DEC, in arcsec from galaxy center
- **ellcoo_r**: the elliptical polar coordinate radial offset, from galaxy center, R in arcsec
- **ellcoo_rre**: the elliptical polar coordinate radial offset, from galaxy center, as R/Re
- **ellcoo_rkpc**: the elliptical polar coordinate radial offset, from galaxy center, R in h-1 kpc
- **ellcoo_theta**: the elliptical polar coordinate angle offset, from galaxy center, theta in degrees
- **skycoo_units**: the physical units of skycoo_x/y
- **ellcoo_r_units**: the physical units of ellcoo_r
- **ellcoo_rre_units**: the physical units of ellcoo_rre
- **ellcoo_rkpc_units**: the physical units of ellcoo_rkpc
- **ellcoo_theta_units**: the physical units of ellcoo_theta

Each **images** item has the following features:

- **filter**: the image band or filter, e.g. griz
- **array**: the 2d array pixel values
- **array_units**: the physical units of the data
- **psf**: the reconstructed PSF pixel values
- **psf_units**: the physical units of the data
- **scale**: the image pixel scale
- **scale_units**: the units of the image scale

Each **maps** item has the following features:

- **group**: the original MaNGA DAP extension, e.g. 'emline_gflux'
- **label**: the full map label, made from the HDU extension name + channel, e.g. 'emline_gflux_ha_6564'.
- **array**: the 2d array of pixel values
- **ivar**: the 2d array of inverse variance values
- **mask**: the 2d array of pixel quality mask values
- **array_units**: the physical units of the map data

**Collection Process**   We use the publicly accessible data for MaNGA from the SDSS-IV 17th data release (DR17)[29].

**Preprocessing/cleaning**   During processing all IFU cubes, and maps have been resized to the same spatial grid of 96 x 96, with zero-padded elements added around the edges of the data. Thus all cubes have shape 96 x 96 x 4563 (42,882,048 pixels), and all maps have shape 96 x 96 (9,216 pixels).

---

[29]https://www.sdss4.org/dr17/manga/

**Uses**    The public MaNGA DR17 dataset has been used in a number of scientific publications in extra-galactic astronomy, as well as in machine learning specific applications.

**Acknowledgements and Terms of Use**    All SDSS data released in our public data releases is considered in the public domain. When using the data included in this compilation, the following acknowledgements should be included:

Funding for the Sloan Digital Sky Survey IV has been provided by the Alfred P. Sloan Foundation, the U.S. Department of Energy Office of Science, and the Participating Institutions. SDSS-IV acknowledges support and resources from the Center for High Performance Computing at the University of Utah. The SDSS website is www.sdss.org.

SDSS-IV is managed by the Astrophysical Research Consortium for the Participating Institutions of the SDSS Collaboration including the Brazilian Participation Group, the Carnegie Institution for Science, Carnegie Mellon University, Center for Astrophysics | Harvard & Smithsonian, the Chilean Participation Group, the French Participation Group, Instituto de Astrofísica de Canarias, The Johns Hopkins University, Kavli Institute for the Physics and Mathematics of the Universe (IPMU) / University of Tokyo, the Korean Participation Group, Lawrence Berkeley National Laboratory, Leibniz Institut für Astrophysik Potsdam (AIP), Max-Planck-Institut für Astronomie (MPIA Heidelberg), Max-Planck-Institut für Astrophysik (MPA Garching), Max-Planck-Institut für Extraterrestrische Physik (MPE), National Astronomical Observatories of China, New Mexico State University, New York University, University of Notre Dame, Observatário Nacional / MCTI, The Ohio State University, Pennsylvania State University, Shanghai Astronomical Observatory, United Kingdom Participation Group, Universidad Nacional Autónoma de México, University of Arizona, University of Colorado Boulder, University of Oxford, University of Portsmouth, University of Utah, University of Virginia, University of Washington, University of Wisconsin, Vanderbilt University, and Yale University.

## A.7    Tabular Datasets

### A.7.1    *Gaia* DR3: `gaia` ☁️ 🐙 🤗 🥐

**Description**    The *Gaia* mission is primarily intended for very precisely measuring astrometry of stars in the Milky Way (MW) from space at an unprecedented scale. Data Release 3 (DR3), in addition to astrometric measurements (position on the sky, parallaxes, proper motions, as well as associated uncertainties), also provides, for example: photometry; physical parameters (stellar surface properties like surface gravity and effective temperature); epoch photometry for variables; orbital elements for binary systems; high-resolution spectra from the radial velocity (RV) spectrometer and low-resolution spectra from the blue and red photometers (BP and RP).

**Composition**    The dataset currently contains the following:

- Low-resolution optical spectra [38]: the spectra are stored as 110 coefficients of Gauss-Hermite polynomials (55 for BP, and 55 for RP). These can be sampled to an arbitrary wavelength grid using the GaiaXPy tool[30].

- Astrometry: on-sky position, parallax, and proper motions, along with the full covariance matrix.

- Photometry: mean $G$, $G_{BP}$, and $G_{RP}$ magnitudes and fluxes, along with errors.

- Photometrically-estimated parameters: stellar parameters (e.g., distance, surface gravity, metallicity, surface temperature) and lower/upper bounds derived from BP/RP spectra, apparent $G$ magnitudes, and parallax.

- Radial velocities: RVs, errors, and stellar parameters of the template used to derive the RVs.

- Corrections and flags: commonly used variables used to make, for example, radial velocity and parallax zero-point corrections.

---

[30]https://www.cosmos.esa.int/web/gaia/gaiaxpy

**Collection Process**   We use the publicly accessible DR3 data through the mirror hosted by the Flatiron Institute[31].

**Preprocessing/cleaning**   We select the subset of 220M stars in DR3 for which *Gaia* mean BP/RP spectra are provided. For these stars, we provide the contents described above. In the future we will include the entire dataset of the nearly 2B sources measured in DR3.

**Uses**   DR3 data has already been used extensively in MW science and in machine learning applications. For example, using DR3, [57] explored and identified non-axisymmetric features in the disc, [58] have created a detailed chemodynamical map of the MW, and [94] have trained a transformer-based model that can generate stellar spectra, estimate physical parameters, and inpaint unobserved spectral regions.

**Acknowledgements and Terms of Use**   This work has made use of data from the European Space Agency (ESA) mission *Gaia* (https://www.cosmos.esa.int/gaia), processed by the *Gaia* Data Processing and Analysis Consortium (DPAC, https://www.cosmos.esa.int/web/gaia/dpac/consortium). Funding for the DPAC has been provided by national institutions, in particular the institutions participating in the *Gaia* Multilateral Agreement. The Gaia data are open and free to use, provided credit is given to 'ESA/Gaia/DPAC'. If you use Gaia DR3 data in your research, please acknowledge it as above.

### A.7.2   Galaxy10 DECaLS gz10 ☁️ 🎧 🤗 🥐

**Description**   Galaxy Zoo (https://www.zooniverse.org/projects/zookeeper/galaxy-zoo/) is a long running citizen science (crowd-sourcing) effort recruiting volunteers to annotate images of galaxies online according to the galaxies structure such as whether or not they have spiral arms or bulges [99]. These structures are useful, and traditionally expensive, tracers of underlying physics such as galaxy evolution [124]. Galaxy10 DECaLS [93] represents a subset of the Galaxy Zoo data, where the original decision tree for Galaxy Zoo is split into ten classes described below.

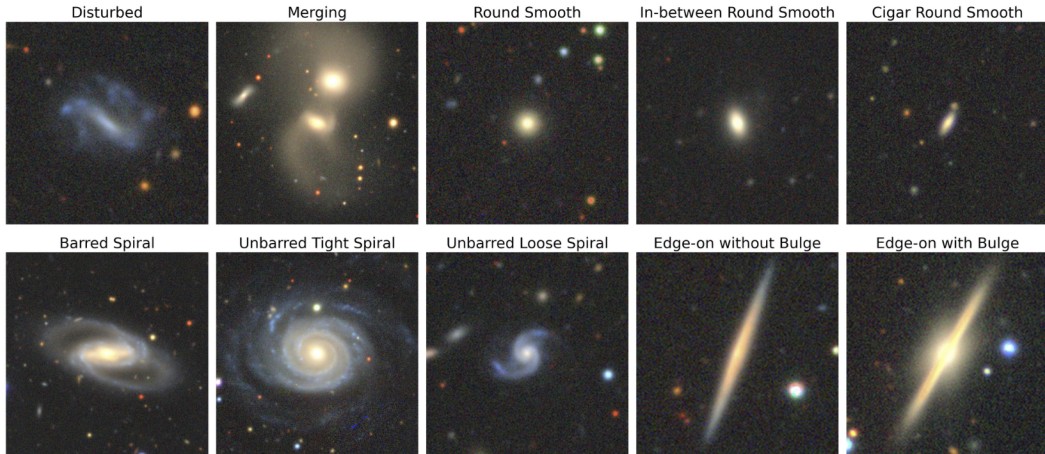

Figure 7: Examples of each class of the GalaxyZoo10 DECaLS data set with their RGB composite image. Adapted from original figure from [93].

**Composition**   [92] The data set is comprised of the images which volunteers saw and an aggregation of their resulting classifications into 10 subsets class labels for the objects spanning 17,736 samples split among the following classes:

1. Disturbed Galaxies (1,081)

---

[31]https://sdsc-users.flatironinstitute.org/~gaia/dr3/

2. Merging Galaxies (1,853)

3. Round Smooth Galaxies (2,645)

4. In-between Round Smooth Galaxies (2,027)

5. Cigar Shaped Smooth Galaxies (334)

6. Barred Spiral Galaxies (2,043)

7. Unbarred Tight Spiral Galaxies (1,829)

8. Unbarred Loose Spiral Galaxies (2,628)

9. Edge-on Galaxies without Bulge (1,423)

10. Edge-on Galaxies with Bulge (1,873)

The data can be loaded in two broad configurations: Firstly, as just the labels with auxiliary data, or the labels with the respective `uint8` PNGs (3x256x256). The auxiliary data includes the right ascension `ra`, declination `dec`, `redshift`, and Multimodal Universe specific object id `object_id`. Additionally, if the images are loaded, the pixel scale of the images is included as `rgb_pixel_scale`.

**Collection Process**   The images presented in this dataset are RGB composites [145] of data collected by the DESI legacy imaging survey [44] and are analogous to what the volunteers were shown while assigning the labels to the galaxies. The collection process for the labels of the underlying data are detailed in [145, 97].

**Preprocessing/cleaning**   There are a number of selection effects made on what data received labels, see [145]. The labels, constructed from the original decision tree (https://data.galaxyzoo.org/gz_trees/gz_trees.html) are collated to select confident and clearly distinguishable classes [92] for simplicity. Only confident labels are maintained as this dataset was intended as a benchmark dataset with minimal or no label noise. Several baseline models for this benchmark are provided, see subsection B.1 Additionally, the images are not the full scientific measurement of the respective object, as done by DESI [44], but rather a RGB composite designed to highlight the respective structure. These images are what the volunteers would have seen whilst labelling the respective objects.

**Uses**   A number of publications have made use of this data, presented on the homepage (https://astronn.readthedocs.io/en/latest/galaxy10.html#some-papers-that-used-galaxy-10). These efforts are largely bench marking the efficacy of various approaches for galaxy morphology classification, as the labels in this dataset are clean and relatively simple in comparison to the raw labels of the decision trees, such as those presented in [145].

**Acknowledgements and Terms of Use**   Galaxy Zoo is described in [99], the GalaxyZoo Data Release 2 is described in [98], Galaxy Zoo DECaLS Campaign is described in Galaxy Zoo is described in [99], the GalaxyZoo Data Release 2 is described in [98], Galaxy Zoo DECaLS Campaign is described in [145], DESI Legacy Imaging Surveys is described in [42].

The Legacy Surveys consist of three individual and complementary projects: the Dark Energy Camera Legacy Survey (DECaLS; Proposal ID #2014B-0404; PIs: David Schlegel and Arjun Dey), the Beijing-Arizona Sky Survey (BASS; NOAO Prop. ID #2015A-0801; PIs: Zhou Xu and Xiaohui Fan), and the Mayall z-band Legacy Survey (MzLS; Prop. ID #2016A-0453; PI: Arjun Dey). DECaLS, BASS and MzLS together include data obtained, respectively, at the Blanco telescope, Cerro Tololo Inter-American Observatory, NSF's NOIRLab; the Bok telescope, Steward Observatory, University of Arizona; and the Mayall telescope, Kitt Peak National Observatory, NOIRLab. The Legacy Surveys project is honored to be permitted to conduct astronomical research on Iolkam Du'ag (Kitt Peak), a mountain with particular significance to the Tohono O'odham Nation.

### A.7.3   PROVABGS `desi_provabgs` ☁️ 🐙 🤗 🦋

**Description**   The PRObabilistic Value-Added Bright Galaxy Survey (PROVABGS) [65] dataset provides measurements of galaxy properties, such as log stellar mass ($\log M_*$), star

formation rate ($SFR$), mass-weighted stellar metallicity ($Z_{MW}$), and mass-weighted stellar age ($t_{age,MW}$) for galaxy spectra in the DESI Bright Galaxy Survey (BGS). These are inferred using a full Bayesian inference framework and state-of-the-art Spectral Energy Distribution (SED) modeling of the DESI spectra. At present, as the DESI ruvey is still on-going, the PROVABGS catalog reports these properties for the early data release (EDR) of the DESI sample.

**Composition**  The best-fit parameters of the PROVABGS SED modeling for $\log M_*$, $SFR$, $Z_{MW}$, $t_{age,MW}$. Additionally, it also includes the $100 \times 13$ array of 100 samples drawn from the 13-dimensional PROVABGS posterior. The 13 parameters correspond to $\log(M_{\mathrm{form}})$[32], $\beta_1$, $\beta_2$, $\beta_3$, $\beta_4$, $f_{\mathrm{burst}}$, $t_{\mathrm{burst}}$, $\gamma_1$, $\gamma_2$, $\tau_{\mathrm{BC}}$, $\tau_{\mathrm{ISM}}$, $n_{\mathrm{dust}}$, $f_{\mathrm{fiber}}$, from which the above properties are derived. We refer the reader to the original PROVABGS documentation[33] for the full details. Overall, the PROVABGS modeling of the DESI EDR contains physical properties for roughly $221,000$ galaxy spectra.

**Collection Process**  We use the reported PROVABGS posteriors from the publicly available DESI EDR BGS PROVABGS catalog[34].

**Preprocessing/cleaning**  We extract the best fit parameters of the PROVABGS SED modeling for $\log M_*$, $SFR$, $Z_{MW}$, $t_{age,MW}$ using the PROVABGS SPS model with non-parametric star formation and metallicity histories and flexible dust attenuation model.

**Uses**  The PROVABGS catalog has been used for physical property estimation from both images and spectra in both self-supervised learning and supervised learning contexts [112].

**Acknowledgements and Terms of Use**  All of the data included in this dataset is of the public domain. We are grateful to the PROVABGS team for making the data available to the community. This research used data obtained with the Dark Energy Spectroscopic Instrument (DESI). DESI construction and operations is managed by the Lawrence Berkeley National Laboratory. This material is based upon work supported by the U.S. Department of Energy, Office of Science, Office of High-Energy Physics, under Contract No. DE–AC02–05CH11231, and by the National Energy Research Scientific Computing Center, a DOE Office of Science User Facility under the same contract. Additional support for DESI was provided by the U.S. National Science Foundation (NSF), Division of Astronomical Sciences under Contract No. AST-0950945 to the NSF's National Optical-Infrared Astronomy Research Laboratory; the Science and Technology Facilities Council of the United Kingdom; the Gordon and Betty Moore Foundation; the Heising-Simons Foundation; the French Alternative Energies and Atomic Energy Commission (CEA); the National Council of Science and Technology of Mexico (CONACYT); the Ministry of Science and Innovation of Spain (MICINN), and by the DESI Member Institutions: www.desi.lbl.gov/collaborating-institutions. The DESI collaboration is honored to be permitted to conduct scientific research on Iolkam Du'ag (Kitt Peak), a mountain with particular significance to the Tohono O'odham Nation. Any opinions, findings, and conclusions or recommendations expressed in this material are those of the author(s) and do not necessarily reflect the views of the U.S. National Science Foundation, the U.S. Department of Energy, or any of the listed funding agencies.

## A.8  Maintenance

To ensure the continued evolution and relevance of MULTIMODAL UNIVERSE, the project is organized as an open and collaborative effort. Contributions from researchers are actively encouraged, with a well-defined process for submitting new data, benchmarks, and improvements via GitHub, as described in the project's CONTRIBUTING document. A dedicated team of maintainers oversees the integration of these contributions, ensuring that the dataset remains up-to-date and continues to meet the needs of the scientific and machine learning communities. This framework will enable regular updates to incorporate new data from

---

[32]Note that $\log(M_{\mathrm{form}})$ is different from $\log$(stellar mass).

[33]https://data.desi.lbl.gov/doc/releases/edr/vac/provabgs/

[34]https://data.desi.lbl.gov/public/edr/vac/edr/provabgs/v1.0/README.md

ongoing and future astronomical surveys, as well as periodic evaluations and improvements based on feedback from the research community.

## A.9 Author statement

The authors of this scientific paper bear full responsibility for any violation of rights that may arise from the collection of the data included in this research.

# B  Details on Benchmark tasks

## B.1  Galaxy10 DECaLS Morphology Classification

For the Galaxy10 DECaLS morphology classification baselines, we train all models using an 80/20 train-test split on the GalaxyZoo10 DECaLS classification labels and the corresponding Legacy Survey RGB-converted images. We train three canonical computer vision baseline networks - ResNet-18 [66], EfficientNetB0 [136], and DenseNet121 [75] - to classify these images. As they are already range compressed in their conversion to RGB images, we do not apply any range compression. To prevent overfitting, we use data augmentation: random horizontal and vertical flips, random gaussian blurring, random affine translation, and color jittering[35]. We train all models using a cross-entropy loss on the training dataset with the Adam optimizer [84]. Training is performed for 10 epochs on a single A100 GPU for each model with a batch size of 128 and a learning rate of $\lambda = 10^{-4}$. We report the Top-1 accuracy on the held-out validation test dataset.

## B.2  Physical Property Estimation

For the physical property inference baselines, we train the image and spectrum models using an 80/20 train-test split on the cross-match between the PROVABGS dataset and the Legacy Survey imaging datasets and the DESI spectrum datasets respectively.

For the image models, we use canonical computer vision baseline networks - ResNet-18 [66], EfficientNetB0 [136], and DenseNet121 [75] - to regress the PROVABGS galaxy properties from the Legacy Survey images. Due to the high dynamic range of the images, we use a range compression scheme to reduce the dynamic range for better optimization. Specifically, we apply *arcsinh* range compression and some additional clamping to the images. We also use data augmentation - random horizontal and vertical rotations and random gaussian blurring - to reduce overfitting. For the spectrum model, we use a convolutional+attention network with the same architecture as the encoder from the state-of-the-art spectrum encoder from [104]. For the photometry model, we use a simple 5-layer MLP with $d = 512$ hidden dimension. Additionally, we $Z$-score the photometry to normalize the data.

We train all models using a mean-squared error (MSE) loss on the training dataset with the Adam optimizer [84]. We $Z$-score the PROVABGS galaxy properties. Additionally, we convert metallicity ($Z_{MW}$) to log metallicity and star-formation rate ($SFR$) to $sSFR = \log SFR - \log Z_{MW}$. We train all models for 25 epochs on a single A100 GPU with a batch size of 256 and a learning rate of $\lambda = 10^{-4}$. We report the $R^2$ performance of all models on the held-out test datasets.

## B.3  PLAsTiCC Photometric Classification and Redshift Estimation

For the PLAsTiCC baseline model, we use an encoder-only Informer model [154] with 8 encoder layers of 12 attention heads each. The model hidden dimension was chosen to be 768 and the layer MLPs have hidden dimension 256. Due to the 2-dimensional position data (each element of the time-series has an associated time and photometric band/wavelength) and irregular sampling of our dataset, we train a positional encoding based on learnable Fourier features following [95]. We also select a random window of length 300 from each example (and zero-pad examples with fewer than 300 observations) to produce inputs of uniform shape. We first pretrain with a masked autoencoding objective:

$$\mathcal{L}_{\mathrm{MAE}}(\phi) = \mathbb{E}_{x \sim P_U, x' \sim \mathcal{A}_{\mathrm{pre}}(\cdot|x)}[(\phi(x') - x)^2] \tag{1}$$

Here, $P_U$ is the distribution of unlabeled data that we pretrain with, $\mathcal{A}_{\mathrm{pre}}$ is the distribution of augmented (masked) inputs, and $\phi$ is the model. We perform pretraining by randomly masking 60% of observations of each time-series, a batch size of 256, and learning rate 1e-4

---

[35]In other astrophysical contexts color jittering tends to be avoided due to its potential to corrupt valuable astrophysical information present in the galaxy colors; however, as we are only concerned with galaxy morphology in this application, we apply it here.

(selected from 1e-3 ∼ 1e-6) for 75,000 steps. We finetune the pretrained model with linear probing for 20,000 steps and learning rate 1e-4, then fine-tuning for 10,000 steps at learning rate of 4e-5. The redshift estimation task uses LP learning rate of 5e-4 and FT learning rate of 1e-4. We decrease the learning rate per step with a linear scheduler.

## B.4 YSE Photometric classification

We demonstrate an implementation of the `snmachine` package [100, 8] on the YSE data set with classes mapped to SNe Ia, SNe II, SNe Ib/c, or other. `snmachine` offers feature extraction methods based on SN Ia templates [64], parametric models [108, 82], and wavelet analysis [90], along with wrappers for several classifiers from `scikit-learn` [113]. We find that wavelet-based feature extraction combined with the Random Forest classifier, with or without the ensemble AdaBoost classifier, achieves a AUC of 0.90in SN Ia one-vs-rest classification.

## B.5 BTSbot Candidate Identification

We report on the performance of BTSbot - a multimodal convolutional neural network which combines images of potential new transient detections with associated metadata - on the BTSbot training sample [119, 120]. Applied to a validation data set comprising 10 per cent of the entire data set, BTSbot achieves an AUC of 0.985 [120]. Given that the BTSbot training set includes multiple alerts for the same transient objects as they evolve, it is important to ensure that train/test splits are based on unique transient objects rather than just unique transient alerts, as in the public data set. While the overall AUC is an important measure, it is also valuable to consider the performance on an object-by-object basis rather than just considering all the individual alerts to determine how many unique transient objects are being identified. Early classification is also of value for this data set to allow transients to be targeted for further observations, therefore a model which is able to accurately identify real transient candidates as early as possible in their light curves will be of great benefit.

## B.6 Contrastive Image-Spectrum Pretraining

We replicate the results of `AstroCLIP` method [112] using the Multimodal Universe cross-matched DESI and Legacy Surveys. We split the dataset using an 80/20 train-test split. We use the same hyperparameter and training suite as in [112] for training the model on the training suite. We then evaluate model performance by evaluating the zero-shot performance of a $k$-NN algorithm with $k = 16$ nearest neighbors on the cross-matched DESI and Legacy Survey and PROVABGS dataset. We report the $R^2$ performance of the model on the held-out test set. Note that the model has never seen the test set during either pretraining or zero-shot "training".

## B.7 From MaNGA Hyperspectral Datacubes to Physical Parameter Images

We demonstrate how 3D hyperspectral datacubes of galaxies can be used to infer 2D image "maps" corresponding to physical parameters, a process typically performed through complete spectrum analysis at each spaxel. In this use case, we train the canonical computer vision baseline network, U-Net [122], on a subset of ∼ 1100 galaxy datacubes from the MaNGA dataset (with a shape of 96 x 96 x 4563) to infer the stellar velocity dispersion map of each galaxy (96 x 96). We split the dataset using an 90/10 train-test split.

The model is trained using a mean absolute error (MAE) loss on the training dataset with the Adam optimizer [84] for 10 epochs. The $R^2$ performance of the model on the held-out test dataset is 0.78, which drops to 0.35 when zero pixels are not considered in the calculation. The reconstruction of the 2D stellar velocity dispersion map for four random galaxies from the test set is shown in Figure 8.

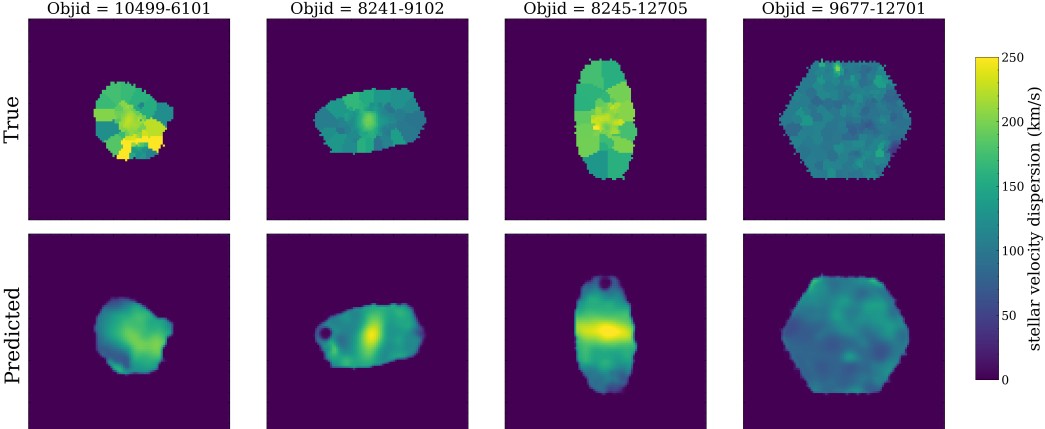

Figure 8: Reconstruction of the stellar velocity dispersion maps by the U-net using the hyperspectral datacubes as input for four galaxies from the MaNGA test set.

## C  Infrastructure

To handle data at the 100TB scale, key decisions around data storage were made to accommodate both I/O and flexibility requirements. The data is stored in HDF5 files for consistency and performance. To allow efficient data access across various file systems, the samples are partitioned into HEALPix[36] regions. This enables simple spatial indexing for objects within similar regions (constrained by the grid size). This partitioning also allows downstream users to download only samples pertaining to specific regions or to match samples across different MULTIMODAL UNIVERSE modalities or datasets. Finally, the data is chunked into HDF5 files with a variable number of samples to prevent individual files from becoming too large.

### C.1  Cross Matching Utility

The MULTIMODAL UNIVERSE framework provides utilities for cross-matching between different astronomical datasets. Below we demonstrate the simplicity of performing cross-matching operations using our provided tools:

```
1 from datasets import load_dataset_builder
2 from mmu.utils import cross_match_datasets
3
4 # Load the dataset descriptions from local copy of the data
5 sdss = load_dataset_builder("MultimodalUniverse/sdss",
6                             trust_remote_code=True)
7 hsc = load_dataset_builder("MultimodalUniverse/hsc",
8                            trust_remote_code=True)
9
10 # Use the cross matching utility to return a new HF dataset
11 dset = cross_match_datasets(
12     sdss,  # Left dataset, contains spectroscopy
13     hsc,   # Right dataset, contains imaging data
14     matching_radius=1.0,  # Distance in arcsec
15 )
```

### C.2  Contributing New Datasets

Detailed guidelines for contributing new datasets to the `MultimodalUniverse`[37] framework are provided in the project's `DESIGN.md` and `CONTRIBUTING.md` documentation. These

---

[36]https://healpix.sourceforge.io/
[37]https://github.com/MultimodalUniverse/MultimodalUniverse/

documents are regularly updated with new tutorials and examples to facilitate community contributions.

For the most up-to-date examples and tutorials, users can refer to our getting started notebook[38].

---

[38]https://colab.research.google.com/github/MultimodalUniverse/MultimodalUniverse/blob/main/notebooks/getting_started.ipynb

