# OpenReview forum: "The Multimodal Universe: Enabling Large-Scale Machine Learning with 100 TB of Astronomical Scientific Data"
_NeurIPS.cc/2024/Datasets_and_Benchmarks_Track — NeurIPS 2024 Track Datasets and Benchmarks Poster_

### Official Review · Reviewer_iM3x · 2024-06-24
**Review of "The Multimodal Universe: Enabling Large-Scale Machine Learning with 70TBs of Astronomical Scientific Data"**

**Rating:** 7
**Confidence:** 3
**Clarity:** The paper is well written.

**Review:**

The article and the contribution are interesting. More specifically, I believe that the "Multimodal Universe" dataset can be useful in accelerating scientific discovery in astronomy.

I have a few concerns:

1. Data Curation Strategy. The authors write "our data curation strategy involves adopting the HuggingFace Datasets framework and imposing a standardized format for each modality": I agree on the approach, but I think the authors should describe more concretely how they "impose a standardized format for each modality": in my experience imposing standardized formats is always difficult. I wonder how the authors enforced this, and how they plan to enforcing it for future extensions of their dataset.

2. Multimodal Cross-Matching. The authors write " offering a set of common cross-matched datasets and utilities that enable end-users to generate custom cross-matches tailored to their specific scientific applications". I think the authors should describe how these utilities work, how end-users can use them, and how difficult it is to extend them to incorporate future extensions of the "Multimodal Universe" datasets.

3. Examples of Machine Learning Tasks on Different Modalities. I do not understand if (a) the example ML tasks were not possible prior to the introduction of the "Multimodal Universe", of if (b) the "Multimodal Universe" enables the development of ML models that yield better predictions in the example tasks. if (a), then the authors should describe why these ML tasks were not possible before. If (b), then the authors should provide comparison with ML models that do not use the "Multimodal Universe".

---

After reading the authors' rebuttal, I change my rating from 5 to 7.

**Strengths:**

- Paper is well written
- Contribution is interesting and potentially very useful to accelerate scientific discovery in astronomy using Machine Learning or Deep Learning
- Dataset is hosted in full at the Flatiron Institute
- HugginFace contains previews of the dataset
- Github repository with script used to create the dataset, and apparently a good community of maintainers.

**Additional Feedback:**

- Abstract: "All codes used to compile the Multimodal ...": I think this should be "All code used to compile the Multimodal..."

- Examples of Machine Learning Tasks on Different Modalities: please use reference to the table with the results, rather than writing "We
report the R2 value on the held-out test set below..." or "We report the performance of BTSbot..." or "We provide baseline photometric classification result..."

**Correctness:**

I think the dataset is constructed in a sound way. I just would like to see some clarification about the process, and more specifically how authors/maintainers impose/ensure standardize data formats.

**Documentation:**

The github repository provide details on the dataset, and there seems to be an active community of contributors and maintainers. The paper describes intended uses of the proposed dataset.

**Ethics:**

I cannot think of any ethical concern.

**Limitations:**

The paper has a "Limitations" section.

I wonder if there are more limitations in terms of difficulty in maintaining/extending the proposed cross-matching utilities when/if the "Multimodal Universe" dataset is extended with more data in the future.

**Opportunities For Improvement:**

See the 3 points above, which I briefly summarize here:

1. Data Curation Strategy: describe how you impose/ensure standardized data formats

2. Multimodal Cross-Matching: describe cross-matching utilities.

3. Examples of Machine Learning Tasks on Different Modalities: clarify the role of the proposed datasets in the example ML tasks.

**Relation To Prior Work:**

Prior work is discussed. I am not an expert in astronomical data, and therefore I cannot judge on the completeness of the prior work section.

**Summary And Contributions:**

The article describes the "Multimodal Universe", a multimodal dataset derived from various existing astronomical scientific datasets. The authors provide a preview of the "Multimodla Universe" datasets on HuggingFace, and the code used to build such datasets is available on Github. The code available on Github also include utilities to generate customized multi-modal datasets by cross matching the existing ones. The article summarizes the characteristics of the data and the corresponding modalities, and illustrates potential uses in machine learning tasks, including some experimental results.

---

> ### Author Rebuttal · Authors · 2024-08-17
>
> We thank the reviewer for this feedback and aim to address the specific points of the review below.
>
> >Data curation strategy:
> > I think the authors should describe more concretely how they "impose a standardized format for each modality"... I wonder how the authors enforced this, and how they plan to enforcing it for future extensions of their dataset.
>
>  The format, or more precisely, the data schema we have defined for each modality resulted from discussions between domain experts representing different science cases and instruments but shared modality (e.g. images, or spectra). We identified the lowest common denominator needed across these use cases to describe the data and defined a number of fields that must be defined for each modality. These fields being fundamental, they are conventionally accessible in any dataset, but may be called differently or not always presented in the same way.
>
> Once the schema for a modality is defined, we enforce standardization of the data at the level of the dataset preparation script that contributors submit through Pull Request. For their submission to be valid, their dataset file must define at a minimum all the fields required for a given modality, and optionally provide additional information. Compliance is currently enforced during the review of the Pull Request.
>
> We provide documentation on the schema we adopt for each modality in our DESIGN.md document [1] and provide contribution guidelines in our CONTRIBUTING.md document on GitHub.
>
> [1] https://github.com/MultimodalUniverse/MultimodalUniverse/blob/main/DESIGN.md
> [2] https://github.com/MultimodalUniverse/MultimodalUniverse/blob/main/CONTRIBUTING.md
>
> > Multimodal cross-matching:
> > I think the authors should describe how these utilities work, how end-users can use them, and how difficult it is to extend them to incorporate future extensions of the "Multimodal Universe" datasets.
>
> Thank you very much for that feedback. We indeed want to make it as easy as possible for users to perform multimodal cross-matching. Based on your feedback we have included a section in our “getting started” notebook [3] to demonstrate how easy it is to perform cross-matching:
>
> ```python
> from datasets import load_dataset_builder
> from astropile.utils import cross_match_datasets
>
> # Load the dataset descriptions from local copy of the data
> sdss = load_dataset_builder("MultimodalUniverse/sdss", trust_remote_code=True)
> hsc = load_dataset_builder("MultimodalUniverse/hsc", trust_remote_code=True)
>
> # Use the cross matching utility to return a new HF dataset, the intersection
> # of the parent samples.
> dset = cross_match_datasets(sdss, # Left dataset, contains spectroscopy
>                             hsc,  # Right dataset, contains imaging data
>                             matching_radius=1.0, # Distance in arcsec
>                             )
> ```
> We will include this example as an appendix in the paper. We will also include an additional appendix describing the technical solutions that make this cross-matching possible at the 100TB scale (which includes a spatial partitioning of each dataset into non-overlapping regions of the sky, and choosing file formats enabling fast column-based access to easily retrieve coordinates and random access to quickly retrieve desired examples).
>
> Regarding instructions for how to contribute to the project and add new datasets, we are providing such guidelines in the DESIGN.md and CONTRIBUTING.md documents mentioned above and aim to continuously improve our documentation with tutorials.
>
> [3] https://colab.research.google.com/github/MultimodalUniverse/MultimodalUniverse/blob/main/notebooks/getting_started.ipynb#scrollTo=bgsKcUUVrjV8
>
> > Examples of Machine Learning Tasks on Different Modalities:
> > I do not understand if (a) the example ML tasks were not possible prior to the introduction of the "Multimodal Universe", of if (b) the "Multimodal Universe" enables the development of ML models that yield better predictions in the example tasks. if (a), then the authors should describe why these ML tasks were not possible before. If (b), then the authors should provide comparison with ML models that do not use the "Multimodal Universe".
>
> While ML approaches to these tasks were undertaken prior to the development of the Multimodal Universe, we emphasize that the Multimodal Universe was created with the goal of accelerating new research in ML for astronomy by reducing the barrier to entry and making astronomical datasets easier to work with. The AstroCLIP work [4], described briefly in Section 4.3 of our paper, is an ideal example of the type of research that would benefit greatly from the Multimodal Universe. The AstroCLIP approach involves multimodal cross-matching (identifying images and spectra of the same object) to perform contrastive pretraining, which is a core component of the Multimodal Universe.
> Assembling the initial version of the dataset for AstroCLIP necessitated over a month of work, but can now be done in 5 mins with the cross-matching utility described above.
>
> [4] Parker et al. Astroclip: A cross-modal foundation model for galaxies, 2024.
> Maintenance:
>
> > I wonder if there are more limitations in terms of difficulty in maintaining/extending the proposed cross-matching utilities when/if the "Multimodal Universe" dataset is extended with more data in the future.
>
> We acknowledge the sustained cost to maintaining this dataset, as with any large data set. We are open sourcing the project, including the construction of the constituent components of the dataset to improve reproducibility. Additionally, we have appointed maintainers to act as stewards of the reproducibility and consistency of the data set. We have split these responsibilities across institutes, research groups, and individuals to improve the likelihood of this project becoming a longstanding and core aspect of machine learning and astronomy research.

---

> > ### Author Response · Authors · 2024-08-26
> >
> > Hi, we were wondering if the reviewer has seen our response? We believe we have addressed the reviewer's key points and are eager to communicate further. Thanks again for your time!

---

> > > ### Comment · Reviewer_iM3x · 2024-09-04
> > >
> > > Dear authors,
> > >
> > > apologies for the delay in my reply. I have read your comments, and I think you have addressed all my questions. I have changed my rating from 5 to 7.
> > >
> > > Best Regards

---

### Official Review · Reviewer_JFFL · 2024-07-23

**Rating:** 6
**Confidence:** 3
**Correctness:** Yes, the claims are correct.
**Clarity:** This paper is well written.

**Review:**

This manuscript proposes a multimodal universe, which is a large-scale, systematic benchmark from astronomical field. This manuscript provides a detailed introduction to the scheme for constructing such an astronomical dataset, demonstrates the advantages of the proposed dataset, and offers demonstrations of machine learning tasks using different modalities of data in the Experiment Section. This can effectively promote the integration of machine learning tasks with research in the field of astronomy. After carefully reading this manuscript, I have two main concerns:

- The writing quality and the layout of the figures/tables in this paper still have significant room for improvement. It is recommended that the author make detailed revisions in the next version. Besides, it is necessary to clarify whether the proposed dataset requires quality control. If quality control is needed, what aspects should be inspected? If quality control is not required, what are the reasons? The author is requested to provide a detailed response during the rebuttal period.

- I observe from Table 3 that the result achieved using ResNet18 is better than that using DenseNet121. Please explain this reason. Why do smaller network (ResNet18) achieve better result? Is it because the evaluation set proposed by the authors is not diverse or objective enough?

**Strengths:**

- The proposed dataset is extremely large in volume and encompasses multiple modalities of data.

- Based on the proposed dataset, the authors propose Contrastive Image-Spectrum Pretraining, which is a self-supervised manner and has been verified to be effective.

- The authors provide the detailed supplementary material which shows more examples and implementation details.

**Additional Feedback:**

Please refer to weakness

**Documentation:**

The proposed dataset is available for downloading. Besides, the authors show sufficient details including the proposed evaluation metric and experiments to support reproducibility.

**Ethics:**

No ethical concerns

**Limitations:**

Yes, the authors have discussed the limitations.

**Opportunities For Improvement:**

- Showing the quality control process.

- Improve the quality of writing and the organizational structure of the paper.

**Relation To Prior Work:**

Yes. It is clearly to discuss the differences

**Summary And Contributions:**

- This paper constructs a large-scale multi-modal dataset from scientific astronomical data

- The multi-modal universe covers hundreds of millions of astronomical observations, where the dataset encompasses a rich variety of data tupes including 70TB multi-channel hyper-spectral images,  spectra, multivariate time series.

---

> ### Author Rebuttal · Authors · 2024-08-17
>
> We thank the reviewer for this feedback and we address specific comments below.
>
> > The writing quality and the layout of the figures/tables in this paper still have significant room for improvement. It is recommended that the author make detailed revisions in the next version.
>
> We thank the reviewer for this comment and will be sure to thoroughly review and revise the draft and figures for the camera-ready submission.
>
> > it is necessary to clarify whether the proposed dataset requires quality control. If quality control is needed, what aspects should be inspected? If quality control is not required, what are the reasons?
>
> The Multimodal Universe is a collection of data products released from various astronomical surveys and telescopes that have gone through extensive processing and quality control by the survey collaborations prior to release. We do not include raw data, but science-ready data products that have already undergone calibration and cleaning from the parent surveys. As a result we perform little to no additional filtering on the data. We only apply the recommended data quality selection cuts by the original surveys and by the domain experts contributing datasets during the dataset compilation phase.
>
> We note that errors may still be present in the data, but if so, it means that the same amount of contamination is expected in science samples. From that point of view, it is even beneficial to have training data that is representative of the target samples.
>
>
> > I observe from Table 3 that the result achieved using ResNet18 is better than that using DenseNet121. Please explain this reason. Why do smaller network (ResNet18) achieve better result? Is it because the evaluation set proposed by the authors is not diverse or objective enough?
>
> We thank the reviewer for this observation. We note first that all the models we consider in this benchmark are of similar orders of magnitude in size (ResNet18 has 12M parameters, DenseNet121 has 8M parameters, EfficientNet B0 has 5M parameters), which is motivated by the relatively small size (17k examples) of the supervised training data for the experiment in Table 3.
> Instead of using the canonical imaging data associated with the galaxy morphology labels, we had initially designed the experiment by cross-matching the labels with another imaging survey available in the Multimodal Universe, but upon closer inspection we find that this further reduced the size of the training set and led to scatter in the metrics. It also meant that our metrics were not directly comparable to previous work (because we were not using the same imaging data).
> To make the benchmark more robust and to allow for direct comparison with published works, we have reverted back to using the full classification dataset with its original imaging data, retrained our benchmark, and included additional points of comparison from the recent literature in an updated version of Table 3 (see attached pdf).
> On this dataset, we find that ResNet18 and DenseNet121 achieve similar performance, and are in line with the performance of a ConvNext-nano reported by [1]. We note that we obtain significantly better performance by EfficientNet.

---

> > ### Comment · Reviewer_JFFL · 2024-08-23
> > **Thanks for your rebuttal**
> >
> > Thanks the authors' response, which addresses my concerns regarding quality control and experimental setup. I am considering maintaining my positive score and hope the authors can improve the writing quality in their revision.

---

### Official Review · Reviewer_a5Ep · 2024-07-24
**Thorough Dataset Curation, Potentially Impactful for ML Applications in Astronomy**

**Rating:** 7
**Confidence:** 3
**Correctness:** Yes.
**Clarity:** Yes.

**Review:**

The following points outline the strengths associated with the submission.

1. **Presentation and Clarity**: First, I’d like to note that I thoroughly enjoyed reading the paper. The paper (and supporting details in appendix) are well-written, heavily detailed and are easy to follow. I particularly like how the authors do a great job of outlining the problem setup (dispersed data sources), motivation behind MU, underlying design considerations while curating the dataset (all while recognizing what can and cannot be addressed), subsequent tasks of interest, performance of existing architectures and what can be worked upon from a machine learning standpoint (how to effectively leverage weakly cross-matched samples from different sources). The authors were also upfront in identifying important considerations surrounding limitations (either w.r.t. the data or existing architectures) and availability and maintenance.
2. **Originality**: The novelty of the submission lies in (1) exhaustively collecting astronomical data from several seemingly disparate sources, (2) collating them in a usable format and (3) ensuring routine improvement and maintenance. While this may not seem as much at a first glance, I suspect this will potentially have a substantial impact in making “ML for astronomy” data accessible to domain-specific and general practitioners alike.
3. **Quality and Significance**: The quality of the dataset inherently depends upon the underlying surveys from which data is sourced. On top of this, the quality relies upon the unified format (and the extent to which it can be updated with evolving considerations). In terms of significance, I think the dataset can potentially be quite useful in lowering the barrier-to-entry for “ML for astronomy” research, which can encourage potentially seamless integration of more modern methods for such use-cases. At the time of writing this review, there is already considerable interest along these lines for the dataset (as inferred from the issues and PRs of the associated github repository). Standardizing datasets, and potentially trained models in future iterations will likely be quite useful for this specific sub-domain.

I don’t have any major weaknesses to point out in the paper, except for the following points (addressing which can potentially improve the submission).

1. **Use of Modern Computer Vision Backbones**: The supervised baselines, especially for the image modalities, would benefit from including more modern computer vision architectures (and initializations), that have proven to be quite useful in recent years. In particular, I’d be curious to see how well the transformer (DINOv2, EVA) and hybrid (ConvNext) backbones perform when the input modalities are images. Beyond (perhaps) establishing strong baselines, another reason why I suggest this is due to the relatively low performance on the morphology identification task.
2. **Cross-source Transfer for Tasks**: Since the authors note that there may be differences in the data collection process for different sources (for the same modalities), it might be helpful to check how well models trained for a task on one source perform on another. It might also help gauge any actionable insights for training (broadly) generalizable models on data from MU.
3. **Outlining Additional Desiderata Beyond Performance**: L264-266 highlights how ML systems can face certain issues when trained with closed-world assumptions (train / test IID). It might be helpful to contextualize some of these issues – distribution shift (perhaps as above), uncertainty calibration in context of the data quality and tasks present in MU – since desired considerations for astronomical tasks (say, tolerance for miscalibration) may be different from standard ML settings for computer vision or otherwise.

Overall, I really like the submission (and the proposed collection of datasets). I think it can potentially be quite useful from the ML-astronomy sub-domain.

**Strengths:**

Please refer to the main review for outlined strengths.

**Additional Feedback:**

Please refer to the points under weaknesses and opportunities for improvement.

**Documentation:**

Yes.

**Ethics:**

No.

**Limitations:**

Yes. The authors have adequately potential limitations and considerations surrounding routine maintenance.

**Opportunities For Improvement:**

I think addressing the points outlined under weakness – modern backbones for images, evaluating cross-source transfer, contextualizing robustness, calibration considerations for astronomy tasks would help improve the current submission.

**Relation To Prior Work:**

Yes.

**Summary And Contributions:**

The authors introduce Multimodal Universe (MU), a large-scale dataset of astronomical observations spanning different modalities (images, time-series data and spectral observations), curated from several individual astronomical surveys, collated into unified (open-sourced) format to encourage development of machine learning models for different tasks of interest (morphology classification, property estimation, learning general-purpose embeddings for multimodal astronomical tasks). The authors outline the data creation process – where practitioners across institutions are asked to collate data (with supporting documentation) in a shared format, which is then cross-matched (to the extent possible) and stored in a shared open-sourced library that can be used by interested researchers. Different components of MU are outlined in substantial detail (in the main paper as well as the appendix) – with links to representative subsets allowing a reader to browse samples. The authors demonstrate that it’s possible to use a subset of datasets to train classical and modern deep learning models for morphology estimation, physical property prediction, identifying transient events – tasks which would require domain-specific expert opinions. Additionally, the authors also demonstrate it’s possible to learn representations from paired data in MU that are better suited than the supervised counterparts for some of the tasks. Finally, the authors posit the dataset as a resource to accelerate and democratize machine learning for astronomy and outline important limitations and considerations for future maintenance.

---

> ### Author Rebuttal · Authors · 2024-08-17
>
> Thank you very much for your review and constructive feedback, we aim to address below the different points you raised.
>
> > Use of Modern Computer Vision Backbones:
> The supervised baselines, especially for the image modalities, would benefit from including more modern computer vision architectures (and initializations), that have proven to be quite useful in recent years.
>
> We have robustified and extended our benchmark on galaxy image classification to include results for a ConvNext model, which constitutes a strong baseline from the recent literature in this field. We note that due to the small size of this supervised classification dataset (17k examples), it is difficult to directly train vision transformer models or larger modern architectures.
> Our goal with this benchmark is to motivate larger scale self supervised pre-training on Multimodal Universe data followed by fine tuning. This falls outside of the direct scope of this paper, but to prompt researchers to explore that direction, we also report in a revised version of Table 3 (see attached pdf) results from [1] based on larger scale supervised pretraining that illustrates the significant performance gains that can be unlocked by leveraging pretraining data.
>
> [1] Walmsley, et al. Scaling Laws for Galaxy Images, https://arxiv.org/abs/2404.02973
>
> > Cross-source Transfer for Tasks:
> it might be helpful to check how well models trained for a task on one source perform on another. It might also help gauge any actionable insights for training (broadly) generalizable models on data from MU.
>
> Thank you for this suggestion, it is indeed an important consideration. We have not yet included a new experiment in this direction, but transfer between data surveys is a well documented consideration in the field and we will mention in the text some of the relevant work (e.g. [1,2]). We will also aim to include a meaningful benchmark of cross-source transfer in the near future to the set of benchmarks included in the repository.
>
> We also note that, in contrast to other datasets in the field, we provide all of the necessary metadata for researchers to consider instrumental and survey differences across various data sources. This allows for the development of “universal” models that can make use of that additional information to correctly interpret the data and transfer between surveys. These models are beyond the scope of this paper but we will further insist on this possibility in the text.
>
> [1] Dominguez Sanchez et al, (2019) Transfer learning for galaxy morphology from one survey to another. https://arxiv.org/abs/1807.00807
>
> [2] Ćiprijanović et al, (2023) DeepAstroUDA: Semi-Supervised Universal Domain Adaptation for Cross-Survey Galaxy Morphology Classification and Anomaly Detection. https://arxiv.org/abs/2302.02005
>
> > Outlining Additional Desiderata Beyond Performance:
> It might be helpful to contextualize some of these issues – distribution shift (perhaps as above), uncertainty calibration in context of the data quality and tasks present in MU – since desired considerations for astronomical tasks (say, tolerance for miscalibration) may be different from standard ML settings for computer vision or otherwise.
>
> This is a very good point, space permitting, we will extend this paragraph to give more context for these considerations.
> Some of our target tasks already encompass these issues, and in particular the redshift estimation task (estimating the distance of a galaxy from the observer) is an excellent example. Redshift estimation from images is intrinsically uncertain, meaning that regression is not usually sufficient and full conditional density estimation of redshift given images is warranted. The proper calibration of these predicted distribution is then very dependent on the prior induced by the training set, a slight change of galaxy populations between training and target sets (even under the same survey) will lead to miscalibration of the redshift predictions, which can in turn have a major impact on the downstream science.

---

> > ### Comment · Reviewer_a5Ep · 2024-08-25
> > **Thanks for the detailed response**
> >
> > Apologies for the delayed response to the rebuttal. Thanks to the authors for providing a detailed response to mine as well as other reviewers' concerns.
> >
> > My primary concerns were surrounding 3 key points -- (1) use of modern computer vision backbones, (2) assessing cross-source transfer of trained models and (3) outlining additional task-specific considerations beyond performance. For (1), the authors included results from related work to help provide an anecdotal reference point. For (2), the authors indicated plans to include such experiments in future versions (in the repository). Additionally, for (2) and (3) the authors also mentioned that they will include a discussion. As far as my concerns go, I think the rebuttal sufficiently helps address them.
> >
> > Regarding concerns raised by other reviewers, surrounding quality control, discrepancy in certain comparisons and other curation, maintenance and other clarifications -- I think the author responses do a good job of addressing those as well (as already noted by R-JFFL. Given this, I'd like to maintain my original score of accept for the submission.

---

### Decision · Program_Chairs · 2024-09-26

**Decision:**

Accept (Poster)

**Comment:**

The Multimodal Universe is a large scale dataset comprising astronomical data geared towards machine learning, containing 70TB of multi-channel hyper-spectral images, spectra, multivariate time series. In the current era of large foundation models, the general value of a dataset this complete will be very high to those with the computational resources to work in the area of astronomy. On this merit alone I agree with the authors who unanimously agreed the work warrants acceptance. Furthermore I was happy to see the author's thorough response to requests for further details on the data curation and quality control process.